# SEE: SEE EVERYTHING EVERY TIME - BROADER LIGHT RANGE IMAGE ENHANCEMENT VIA EVENTS

## ABSTRACT

Event cameras, with a high dynamic range exceeding $120dB$, significantly outperform traditional cameras, robustly recording detailed changing information under various lighting conditions, including both low- and high-light situations. However, recent research on utilizing event data has primarily focused on low-light image enhancement, neglecting image enhancement and brightness adjustment across a broader range of lighting conditions, such as normal or high illumination. Based on this, we propose a novel research question: how to employ events to enhance and adjust the brightness of images captured under broader lighting conditions. To investigate this question, we first collected a new dataset, **SEE-600K**, consisting of 610,126 images and corresponding events across 202 scenarios, each featuring an average of four lighting conditions with over a 1000-fold variation in illumination. Subsequently, we propose a framework that effectively utilizes events to smoothly adjust image brightness through the use of prompts. Our framework captures color through sensor patterns, uses cross-attention to model events as a brightness dictionary, and adjusts the image's dynamic range to form a broader light-range representation (BLR), which is then decoded at the pixel level based on the brightness prompt. Experimental results demonstrate that our method not only performs well on the low-light enhancement dataset but also shows robust performance on broader light-range image enhancement using the SEE-600K dataset. Additionally, our approach enables pixel-level brightness adjustment, providing flexibility for post-processing and inspiring more imaging applications.

## 1 INTRODUCTION

Every day, from daylight to nighttime, the illuminance varies from about 100,000 *lux* (bright sunlight) to approximately 0.1 *lux* (starlight) (Koshel, 2012). Maintaining stable imaging under diverse natural lighting conditions is a significant challenge. To achieve this, a series of influential works have emerged, including automatic exposure (Bernacki, 2020), exposures correction (Yuan & Sun, 2012), low-light enhancement (Li et al., 2021) and high dynamic range (HDR) imaging (McCann & Rizzi, 2011). However, traditional cameras are limited by their imaging principle of synchronously capturing intensity values across the entire sensor, with a dynamic range of only 60 to 80 $dB$ Hasinoff et al. (2016); Rebecq et al. (2019). Consequently, these traditional methods find it difficult to capture imaging information under a wide range of lighting conditions at the input (Gehrig & Scaramuzza, 2024; Gallego et al., 2020). If the exposure is inaccurate - over and under exposures - traditional cameras lose the potential to restore images under complex lighting conditions due to limited bits-width and noise. Unlike traditional cameras, event cameras Gallego et al. (2020) asynchronously record pixel-level changes in illumination, outputting the direction of intensity change (positive or negative) at each pixel with extremely high dynamic range (120 $dB$), which far exceeds the capability of traditional cameras in capturing various lighting intensity.

Research leveraging the events for image brightness enhancement can be divided into three categories. **(1) event-based image reconstruction**, which aim to reconstruct images only from events. However, these methods (Rebecq et al., 2019; Stoffregen et al., 2020; Wang et al., 2024) rely solely on events, facing uncertainties during reconstruction, and the events usually contain heavy noise, which leads to color distortion and limited capabilities of generalization. **(2) event-guided HDR imaging** (Cui et al., 2024; Yang et al., 2023; Messikommer et al., 2022) , which aims to employ events to extend the dynamic range of images or video to match human vision. However, synthesizing HDR images as ground truth

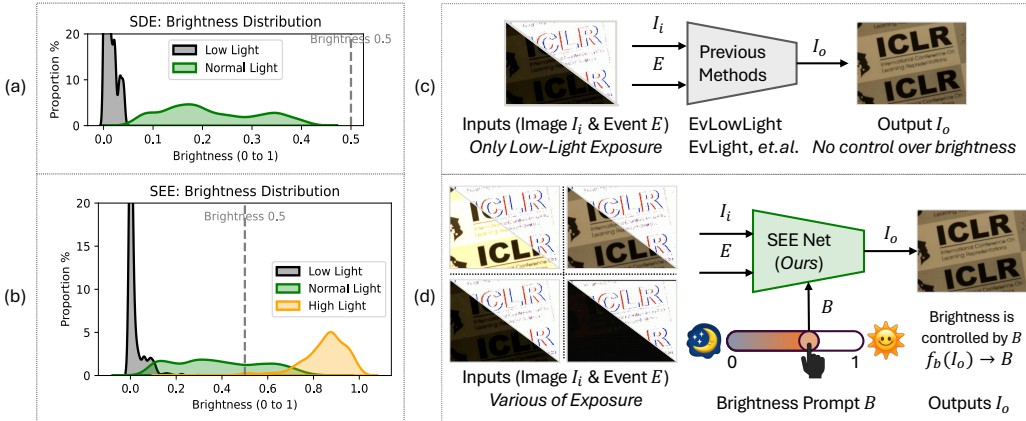

Figure 1: (a) and (b): Brightness distributions of the SDE dataset (0∼0.45, low to normal light) and our SEE-600K dataset (0∼1, a broader light range). (c): Previous methods (Liang et al., 2024; 2023) directly map low-light images to normal-light images. (d): Our SEENet accepts inputs across a broader brightness range and adjusts output brightness through prompts. $f_b$ refers to the function that calculates the brightness of an image.

is difficult. (Cui et al., 2024) introduced the first real-world dataset containing paired color events, low dynamic range, and HDR images, with only includes 1,000 HDR images. (Messikommer et al., 2022) used nine images with different exposures to synthesize an HDR image as the ground truth and utilized multi-exposure frames and events as inputs to generate an HDR image. While HDR imaging aims to expand dynamic range, collecting HDR datasets is difficult, and these methods have not been evaluated for tasks like low-light enhancement or high-light restoration (Tursun et al., 2015; Jayasuriya et al., 2023). **(3) event-guided low-light enhancement** (Liang et al., 2024; 2023; Liu et al., 2023; Jiang et al., 2023), which is designed to adjust low-light images to normal-light conditions through brightness adjustment and noise reduction. Liang et al. (2024) represents the latest research and proposed the first event-based low-light image enhancement dataset, SDE (see Fig. 1 (a)). Prior to this, Liang et al. (2023); Liu et al. (2023); Jiang et al. (2023) explored using motion information from events and employed varying neural networks to improve the mapping from low-light images to normal-light ones, as shown in Fig. 1 (c). However, these strategies only focus on the improvement of mapping ability for low-light inputs, limiting their capacity to adjust brightness across a broader range of lighting conditions, *e.g.*, normal or high-light images. Furthermore, due to the uncertainty in the standard for normal-light image collection—as the normal-light images are relative to low-light images (as shown in Fig.1 (a))—these methods introduce ambiguity during the training process because they can only map low-light images to normal-light ones based on a single set of low- and normal-light data pairs captured per scene. Overall, current research focuses on low-light enhancement, neglecting image enhancement and processing under a wider range of lighting conditions. Therefore, ***how to use events to enhance and adjust the brightness of images across a broader range of lighting conditions*** becomes a more worthwhile research question.

To address this novel research question, we first formulate the imaging model for brightness adjustment (Sec.3) and define the learning task. We aim to perceive lighting information from events, utilizing brightness prompts to convert this lighting information into images with a specific brightness. In doing so, other image quality aspects (like sensor patterns, noise, color bias, and so on) are taken into consideration.

To realize our proposed task, we first collecte a new dataset by emulating each scene in different lighting conditions, covering a broader luminance range (Sec.4), as shown in Fig.1 (b) and (d). By capturing multiple lighting conditions per scene, we enable mappings across diverse illumination scenarios, providing rich data for model training. To tackle the challenges of spatio-temporal alignment of video and event streams under various lighting conditions, we design a temporal alignment strategy relying on programmable robotic arms and inertial measurement unit (IMU) sensors. As a result, we obtain a temporal registration error up to one millisecond and a spatial error at the sub-pixel level (∼ 0.3 *pixel*). Finally, we build a large-scale and well-aligned dataset containing 202 scenes, each with 4 different lighting conditions, summing up to 610,126 images and the corresponding event data. We term this dataset as SEE-600K, which supports learning the mappings among multiple lighting conditions.

Building on the SEE-600K dataset, we propose a compact and efficient framework, SEE-Net, for the proposed new tasks (Sec. 5). An event-aware cross-attention is used to enhance image brightness, and the brightness-related prompt is introduced for controlling the overall brightness. This approach effectively captures and adjusts lighting across a broader range of illumination conditions, providing flexibility and precise control during inference. Despite of the advantage of performance, SEE-Net still remains effective, compact, and lightweight with only **1.9 *M*** parameters.

Our method has been evaluated on two real-world datasets, SDE (Liang et al., 2024) and SEE-600K. Quantitative results demonstrate that our framework fits well to a broader range of lighting conditions (Sec. 6). Furthermore, our framework allows for smooth brightness adjustment, providing precise exposure control. Therefore, this flexibility significantly improves post-processing capabilities and enables potential applications in advanced imaging and processing tasks.

## 2 RELATED WORKS

**Frame-based:** These brightness enhancement methods aim to improve image quality under challenging illumination conditions. Retinexformer (Cai et al., 2023) and other Retinex-based frameworks (Zhang et al., 2021; Wu et al., 2022; Fu et al., 2023) decompose reflectance and illumination with complex training pipelines. Other approaches, *e.g.*, structure-aware models (Xu et al., 2023b; Wang et al., 2023c), utilize edge detection or semantic-aware guidance to achieve sharper and more realistic results. Exposure correction strategies (Afifi et al., 2021; Panetta et al., 2022; Ma et al., 2020) target both overexposed and underexposed areas, leveraging multi-scale networks or perceptual image enhancement frameworks to synthesize correctly exposed images. However, the reliance on RGB frames with limited bit depth, limits the adaptability to dynamic lighting conditions, making it difficult to handle a broader range of lighting scenarios. **Event-based:** These methods focus on reconstructing images or videos exclusively from event data. For instance, Duwek et al. (2021) introduced a two-phase neural network combining CNNs and SNNs, while Pan et al. (2019) proposed the event-based double integral model to generate videos. Stoffregen et al. (2020) enhanced event-based video reconstruction by introducing the new dataset. Additionally, Liu & Dragotti (2023); Wang et al. (2024) developed a model-based deep network to improve reconstructed video quality. However, these event-based approaches face challenges due to event data noise, often leading to color distortion and limited generalization. **Event-guided:** These works are centered on enhancing images captured in low-light conditions. E.g., Zhang et al. (2020) and Liu et al. (2024) recovered lost details in low-light environments by reconstructing grayscale images. Similarly, Liang et al. (2023) and Liu et al. (2023) improved low-light video enhancement by leveraging motion information from events to enhance multi-frame videos and integrating spatiotemporal coherence. Furthermore, Jin et al. (2023) and Jiang et al. (2023) utilized events to recover structural details and reconstruct clear images under near-dark situations. Most notably, Liang et al. (2024) introduced the first large-scale event-guided low-light enhancement dataset, which is significant for the development of this field. While these methods use events for brightness changes and structural recovery in low-light conditions, they are limited to enhance low-light images with single mapping and cannot handle brightness adjustments across a broader range of lighting conditions, including normal- and high-light.

## 3 PRELIMINARIES AND NEW TASK DEFINITION

In this section, we formalize the physical model underlying our approach to enhance and adjust image brightness across a broader range of lighting conditions using events. Imaging is fundamentally the process of capturing the radiance of a scene, represented as a radiance field $L(t)$ varying over a preset slot $t$. The illuminates of light in daily life span a vast range, from 0.1 *lux* (starlight) to $1e6$ *lux* (direct sunlight). The goal of brightness adjustment is to recover or estimate $L(t)$ and tone-map it into an image that is visually suitable for human perception.

Traditional cameras record light signals through exposure (Mendis et al., 1997). This voltage is influenced by the Gaussian noise $N = \mathcal{N}(\mu, \sigma^2)$ ($\mu$ is the mean and $\sigma^2$ is the variance), and the photon shot noise $P = \mathcal{P}(k)$, where $k \propto L(t)$ is the number of photons, proportional to light intensity. In low-light conditions, Gaussian noise dominates, while in high-light conditions, photon shot noise becomes more significant. These noises influence the final value in the RAW image, simply represented as $I_{\text{raw}} \approx \mathcal{Q}(L(t) + P + N)$, where $\mathcal{Q}$ is the quantization function that converts the

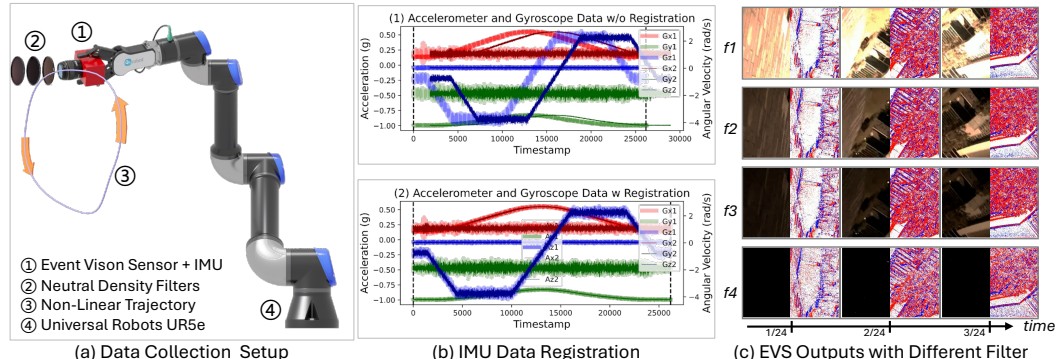

Figure 2: **(a) data collection setup:** Universal Robots UR5e arm replicates precise trajectories with an error margin of $0.03mm$. **(b) IMU data registration:** *b (1)* shows unregistered IMU data, while *b (2)* displays registered data after timestamp alignment. **(c) EVS outputs with different filters:** *f1* to *f4* demonstrate the different ND filters, depicting various lighting levels.

continuous voltage into discrete digital signals, typically ranging from 8 to 12 bits. The shape of the image $I_{raw}$ is $H \times W \times 1$, where $H$ and $W$ are the image resolution. The RAW image is then further processed through image signal processing (ISP) $f_{isp}$, which includes multiple steps *e.g.*, denoising, linear and non-linear transformations, resulting in a RGB image as $I_{rgb} = f_{isp}(I_{raw})$, with the shape of $H \times W \times 3$. An accurate image exposure procedure recovers $I_{rgb}$ corresponding to $L(t)$, up to a high degree meeting the following three characteristics: **(1) accurate exposure**: The mean value of $I_{rgb}$ falls within the range $[0.4, 0.7]$ (Mertens et al., 2009). **(2) noise-free**: The influence of $N$ and $P$ is suppressed to a visual-acceptable level. **(3) color neutrality**: The gray levels calculated from the RGB channels should be consistent (Buchsbaum, 1980). However, traditional cameras sometimes fail to capture sufficient details in extreme-lighting scenes. Under such low-light conditions, images may lack visible details and be contaminated by noise, while in high-light conditions, images may suffer from oversaturation, losing texture and edge information.

Event cameras asynchronously detect illumination changes at each pixel, making them ideal for capturing scenes with extreme or rapidly changing lighting conditions (Gallego et al., 2020). The event stream's outputs are formatted as 4 components: $(x, y)$ (pixel coordinates), $t$ (timestamp), and $p \in \{+1, -1\}$ (polarity, indicating light intensity increase or decrease). Events are triggered when the change in illumination exceeds a threshold $C$ ($\Delta L = \log(L(t)) - \log(L(t - \Delta t))$ where $|\Delta L| > C$). We jointly leverage the complementary information from an image $I_{rgb}$ and its corresponding events $E$ to recover a high-quality well-illuminated image $\hat{I_{rgb}}$ that accurately represents the scene radiance $L(t)$, while also allowing for adjustable brightness. To achieve this, we introduce a brightness prompt $B$ that controls the overall brightness of the output image. This allows us to map the $L(t)$ into an image that is optimally exposed for human observation. Our task setting can thus be formulated as Eq. 1, where $f_{see}$ is our proposed model, as shown in Fig. 1.

$$f_{see}(I_{rgb}, E, B) \rightarrow \hat{I_{rgb}}. \tag{1}$$

This formulation has two advantages: **(1) robust training**: By inserting the brightness prompt $B$ during training, we can decouple the model from biases in the training data with specific brightness level, enabling the model to generalize better over illuminates domain. **(2) flexible inference**: During inference, the prompt $B$ can be set to a default value (*e.g.*, $B = 0.5$) to produce images with general brightness, or be adjusted to achieve different brightness levels, providing flexibility for applications requiring specific exposure adjustments or artistic effects. *Due to space limitations, please refer to the supplementary material for more details of this section.*

## 4 DATASET COLLECTION

In this section, we introduce the SEE-600K dataset, designed to contain *(1) multiple lighting conditions*, *(2) complex motion trajectories* and *(3) spatio-temporal alignment*. Unlike the state-of-the-art SDE dataset (Liang et al., 2024), we capture data across multiple lighting conditions. Most importantly, SEE-600K is nearly 20 times larger than the SDE dataset, providing a stronger foundation for training models with better generalization.

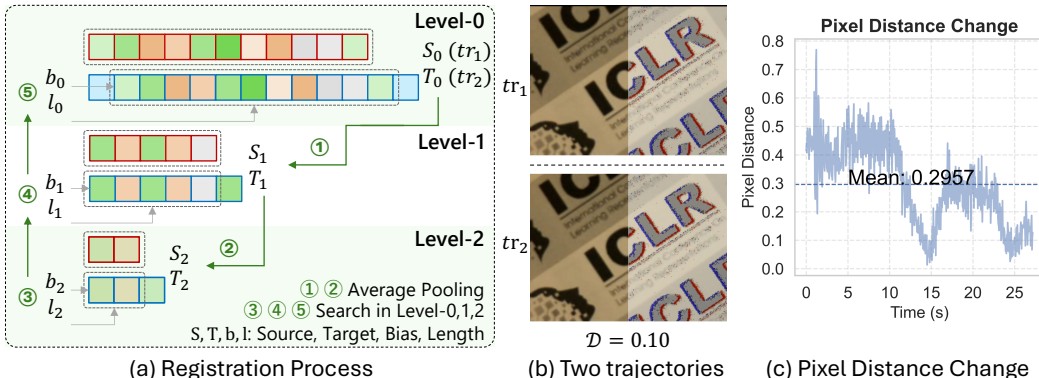

(a) Registration Process      (b) Two trajectories      (c) Pixel Distance Change

Figure 3: **(a) registration process:** Illustration of the multi-level registration process, showing how trajectories, $S$ and $T$, at various levels are iteratively aligned. **(b) two trajectories:** Example of two aligned images captured along two trajectories. **(c) pixel distance change:** Temporal distance of pixel between two registered videos, showing a mean alignment error of 0.2957 pixels over time.

**(1) multiple lighting conditions**: Our approach is based on the principle that lighting transitions continuously from low to high intensity. Unlike previous datasets (Liang et al., 2024; Wang et al., 2021), which captured only a *single* pair of low-light and normal-light conditions, we focus on *multiple* samples. To cover a broader lighting range, we record an average of four videos per scene, using neutral density (ND) filters at three levels ($1/8$, $1/64$, $1/1000$) and one without a filter. We also adjust the aperture and exposure settings to capture each scene under diverse lighting conditions. **(2) complex motion trajectories**: We employ the Universal Robots UR5e robotic arm, which can provid high stability and repeat the same non-linear trajectory with an error margin of **0.03 mm** (Liang et al., 2024; Brey et al., 2024), allowing us to capture multiple videos with spatial consistency, as exhibited in Fig. 3 (a). **(3) spatio-temporal alignment**: While the robotic arm guaranteed spatial alignment, asynchronous control over the camera's start and stop times inevitably introduced timing deviations. To resolve this, we propose an IMU-based temporal alignment algorithm, as shown in Fig. 3 (b). IMU streams synchronized to events and video with microsecond timestamps in the DVS346 camera. Additionally, the IMU stream depends only on motion trajectory and enjoys a temporal resolution of 1000 Hz. Based on this, our algorithm achieves precise temporal alignment, ensuring synchronization across the entire dataset, as displayed in Fig. 3 (c).

**Temporal IMU Registration Algorithm:** We propose an IMU data registration algorithm that aligns the source sequence $S$ and target sequence $T$ by finding the optimal bias $b$ and matching length $l$ to minimize the $L_1$ distance between them. Given the high resolution of IMU data at $1000Hz$, an exhaustive search for the optimal bias is computationally infeasible. To address this, we introduce a multi-level iterative strategy. First, we denoise the IMU data using a Kalman filter (Mirzaei & Roumeliotis, 2008). Then, the average pooling is utilized to reduce the sequences to two additional levels, Level-1 ($S_1$, $T_1$) and Level-2 ($S_2$, $T_2$), as shown in Fig. 3 (a)-①②. This reduces computational complexity while preserving essential alignment features. The window size is chosen based on our video durations, which ranges from 10 to 120 seconds. We perform a coarse search for the optimal bias $b$ and matching length $l$ at the lowest resolution (Level-2). The results from this level serve as center points for finer searches at higher resolutions. Specifically, the bias and length identified at each level guide local searches at the next level up, as displayed in Fig. 3 (a)-③④⑤. At Level-1 and the original data level (Level-0), we only need to search locally around these center points. This hierarchical approach efficiently achieves high matching accuracy with significantly reduced computational effort.

**Spatial-Temporal Alignment Evaluation:** To evaluate the accuracy of our IMU registration algorithm, we capture the same scene twice under identical lighting conditions, as illustrated in Fig. 3 (b). We assess the alignment metric between the two image sequences by calculating the pixel-level distance at the corresponding timestamp. *Alignment Metric*: For each image pair, we extract keypoints using SIFT (Lowe, 2004) and then employ the FLANN matcher (Muja & Lowe, 2009) to find matching keypoints between the two images. Based on these matched keypoints, we compute the affine transformation matrix using RANSAC (Fischler & Bolles, 1981). This transformation is subsequently applied to each pixel, allowing us to calculate the displacement distance for every pixel. Finally, the

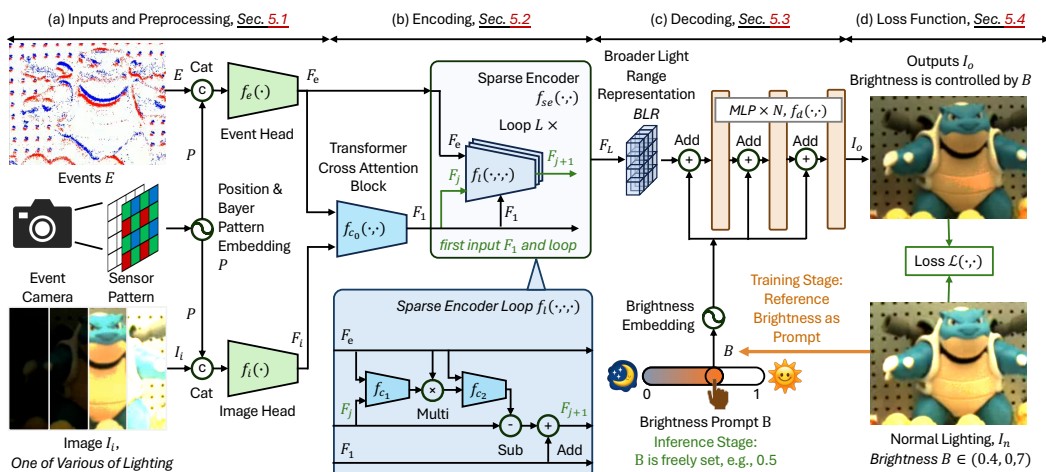

Figure 4: Overview of our proposed framework, called SEE-Net, which is composed of four stages: **(a)** Inputs and Preprocessing, **(b)** Encoding, **(c)** Decoding, and **(d)** Loss Function. This framework takes as input an image captured under a wide range of lighting conditions, along with its corresponding events. The output is a brightness-controllable image, where the brightness is guided by the brightness prompt $B$, enabling flexible pixel-level adjustment during inference.

average pixel distance is employed as the metric for alignment. ***Alignment Results***: In the alignment evaluation, we select scenes with well-defined textures, as illustrated in Fig. 3 (b). After calculating the pixel distances, we observe that the average pixel error between the paired images is $0.2967$ pixels. Throughout the entire time sequence, the pixel-level distance remains below $0.8$ pixels, with the majority of errors being under $0.5$ pixels, as exhibited in Fig. 3 (c). These results demonstrate that the registration accuracy of our dataset reaches sub-pixel precision. *For further details, please refer to the appendix.*

## 5 METHODS

**Overview:** As shown in Fig. 4, our framework, SEE-Net, consists of four implementation parts: **(a)** Inputs and Preprocessing, **(b)** Encoding, **(c)** Decoding, and **(d)** Loss Function. The input is an image $I_i$ and its corresponding events $E$. The output is a brightness-adjustable image $I_o$, where the brightness is controlled by the prompt $B \in (0, 1)$. During training, the brightness prompt $B$ is calculated according to the target image. On the other hand, during testing, $B$ can be freely set, with a default value of 0.5, which follows the exposure control constraint (Mertens et al., 2009; 2007). Overall, the SEE-Net $f_{see}$ can be described by the Eq. 2 to match our learning task in Sec. 3.

$$I_o = f_{\text{see}}(I_i, E, B).\qquad(2)$$

Below, we elaborate the insights and implementation details of each part.

**Inputs and Processing:** This part aims to transform initial inputs into features that retain original information for the encoding stage. The inputs consist of the image $I_i$ and the events $E$, where $I_i$ has a dimension of $H \times W \times 3$ (with $H$ and $W$ representing the height and width, and 3 representing the color channel number). The event stream $E$ is represented as a voxel grid (Tulyakov et al., 2022) with a dimension of $H \times W \times M$, where $M$ represents the number of time slices of events. The events include color information Scheerlinck et al. (2019), which was overlooked in previous works, *e.g.*, (Liang et al., 2024; 2023). Specifically, this DVS346 sensor records events with Bayer Pattern (Lukac et al., 2005). To effectively embed both the color and positional information during framework training, we design the position and bayer pattern embeddings, as shown in Fig. 4 (a). The position and Bayer Pattern are denoted as a vector $(x, y, bp)$, where $x, y$ is the pixel position, and $bp$ denotes the Bayer Pattern index, which takes a value from 0 to 3. We embed this vector into a higher-dimensional feature, termed as $P$, and concatenate it with the inputs. Two layers $1 \times 1$ convolutions, denote $f_e$ and $f_i$, are then applied to obtain the initial event features $F_e$ and image features $F_i$. This process is described by the Eq. 3, where $f_{cat}$ denotes the concatenation function.

$$F_e = f_e(f_{cat}(E, P)), \quad F_i = f_i(f_{cat}(I_i, P)).\qquad(3)$$

**Encoding:** In this stage, we aim to obtain the BLR by employing the event feature $F_e$ to enhance the image feature $F_i$, facilitating noise reduction and the acquisition of broader light range information. Since $F_e$ contains rich information about the lighting changes across different intensity levels, we use it as the source for representing the broader light range. However, event data only records changes in illumination, which differ fundamentally from the static RGB frame modality. This makes directly utilizing event data for broader light representation challenging. To address this, we employ a cross-attention (Liang et al., 2021) for feature fusion, producing the initial fused broad-spectrum feature $F_1$, expressed as $F_1 = f_{c_0}(F_e, F_i)$, where $f_{c_0}$ is a cross-attention block. Then, inspired by previous works (Wang et al., 2020), we utilize sparse learning to generate residuals for $F_1$ from the event features $F_e$. These residuals are progressively generated from the loop that executes $L$ times. Multiple iterations are used because they allow the model to iteratively refine the residuals, capturing finer details and enhancing the feature representations by progressively integrating information from the events. A single loop of this process can be expressed as, $F_{j+1} = f_l(F_e, F_1, F_j)$, where $f_l$ is a loop function that contains two cross-attention blocks as shown in Fig. 4 (b), where $F_j$ and $F_{j+1}$ are the input and output of one loop. After $L$ iterations, the final feature $F_L$ represents the BLR, as described by Eq. 4.

$$F_L = f_{se}(F_e, F_1) = f_l(F_e, F_1, f_l(F_e, F_1, ...f_l(F_e, F_1, F_1))). \tag{4}$$

**Decoding:** The objective of this part is to decode the BLR into a brightness-adjustable image $I_o$. In designing this decoder, we focus on two key insights: (1) The decoding process should be pixel-wise and efficient, allowing for greater flexibility during model deployment; (2) The embedding of the brightness information should be thorough and fully integrated. With these insights, we design the decoder with only a 5-layer MLP as shown in Fig 4 (c). Our decoder begins by encoding the brightness prompt $B \in (0, 1)$ into an embedding vector. To effectively encode the high-frequency brightness prompt into features that are easier for the network to learn (Vaswani, 2017), we introduce a learnable embedding, denoted as $\boldsymbol{B} = f_{pe}(B) = f_{mlp}(f_{cat}(f_{mlp}(B), B))$, which consists of two MLP layers. Through this embedding, the brightness prompt $B$ is transformed into a vector $\boldsymbol{B}$, matching the dimensions of the BLR channels. We then integrate this embedding $\boldsymbol{B}$ into the decoder. To ensure the brightness prompt is fully incorporated and prevent information loss through multiple MLP layers, we employ a multi-step embedding approach, as displayed in Eq. 5, which guarantees that the brightness is progressively embedded throughout the decoding process. During the training phase, the prompt $B$ is derived from the reference image by applying $f_b$ to calculate the global average brightness. In contrast, during the testing phase, $B$ can be set freely, with a typical example being a value of 0.5.

$$I_o = f_d(F_L, \boldsymbol{B}) = f_{mlp}(\boldsymbol{B} + f_{mlp}(\boldsymbol{B} + ...f_{mlp}(\boldsymbol{B} + F_L))). \tag{5}$$

**Loss Function:** The purpose of our loss function is to supervise the prediction $I_o$ using the ground truth $I_t$, with the corresponding brightness $B = f_b(I_t)$. The loss function consists of two main components: image reconstruction loss $\mathcal{L}_i$ and gradient loss $\mathcal{L}_g$. The image reconstruction loss is Charbonnier loss (Lai et al., 2018), which effectively handles both small and large errors. Additionally, we employ gradient loss to improve the structural consistency of the output image. This is achieved by enforcing $L_1$ constraints on the gradients of both the output and ground truth images. Therefore, the overall loss function is formulated as a weighted sum of the image loss and gradient loss, as exhibited in Eq. 6. Here, $\nabla$ denotes the gradient operator, and $\lambda_1$ and $\lambda_2$ are the weights that balance the contributions of two loss terms.

$$\mathcal{L}(I_o, I_t) = \lambda_1 \mathcal{L}_i + \lambda_2 \mathcal{L}_g = \lambda_1 \sqrt{(I_o - I_t)^2 + \epsilon^2} + \lambda_2 \|\nabla I_o - \nabla I_t\|. \tag{6}$$

## 6 EXPERIMENTS

**Experimental Setting:** *Implementation Details:* Our experiments use the Adam optimizer with an initial learning rate of $2e - 4$ for all the experiments. We train our model for 40 epochs on the SDE dataset (Liang et al., 2024). On the SEE-600K dataset, we train for only 20 epochs, as SEE-600K is extremely large. All of our training is conducted on an HPC cluster, with a batch size of 2. To enhance data diversity, we apply random cropping to the images and perform random flips and rotations. *Evaluation Metrics:* We maintain consistency with previous methods (Liang et al., 2024; 2023) by using PSNR and SSIM (Wang et al., 2004). However, since our proposed new problem is highly challenging and most current approaches perform poorly on our SEE-600K dataset, we additionally introduce the $L_1$ distance as a reference.

Table 1: Comparison of different methods on the SDE dataset. The best performances is highlighted in **bold**. †
refers to the original model for the HDR task, which is fine-tuned and trained on SDE

| Method | FLOPs | Params | Events | indoor | | outdoor | | average | |
|---|---|---|---|---|---|---|---|---|---|
| | | | | PSNR | SSIM | PSNR | SSIM | PSNR | SSIM |
| DCE (Guo et al., 2020) | 0.66 | 0.01 | ✗ | 13.91 | 0.2659 | 13.38 | 0.1842 | 13.64 | 0.2250 |
| SNR (Xu et al., 2022) | 26.35 | 4.01 | ✗ | 20.05 | 0.6302 | 22.18 | 0.6611 | 21.12 | 0.6457 |
| Uformer (Wang et al., 2022) | 12.00 | 5.29 | ✗ | 21.09 | 0.7524 | 22.32 | 0.7469 | 21.71 | 0.7497 |
| LLFlow (Wu et al., 2023) | 409.50 | 39.91 | ✗ | 20.92 | 0.6610 | 21.68 | 0.6467 | 21.30 | 0.6539 |
| Retinexformer (Cai et al., 2023) | 15.57 | 1.61 | ✗ | 21.30 | 0.6920 | 22.92 | 0.6834 | 22.11 | 0.6877 |
| E2VID+ (Stoffregen et al., 2020) | 27.99 | 10.71 | ✓ | 15.19 | 0.5891 | 15.01 | 0.5765 | 15.10 | 0.5828 |
| ELIE (Jiang et al., 2023) | 440.32 | 33.36 | ✓ | 19.98 | 0.6168 | 20.69 | 0.6533 | 20.34 | 0.6350 |
| HDRevYang et al. (2023) † | 118.65 | 13.42 | ✓ | 21.13 | 0.6239 | 21.82 | 0.6824 | 21.47 | 0.6531 |
| Wang et al. (2023a) | 170.32 | 7.38 | ✓ | 21.29 | 0.6786 | 22.08 | 0.7052 | 21.68 | 0.6919 |
| eSL-Net (Wang et al., 2020) | 560.94 | 0.56 | ✓ | 21.25 | 0.7277 | 22.42 | 0.7187 | 21.84 | 0.7232 |
| Liu et al. (2023) | 44.71 | 47.06 | ✓ | 21.79 | 0.7051 | 23.35 | 0.6895 | 22.57 | 0.6973 |
| EvLowlight (Liang et al., 2023) | 524.95 | 15.49 | ✓ | 20.57 | 0.6217 | 20.04 | 0.6485 | 20.31 | 0.6351 |
| EvLight (Liang et al., 2024) | 180.90 | 22.73 | ✓ | *22.44* | *0.7697* | *23.21* | *0.7505* | *22.83* | *0.7601* |
| SEENet (Ours) | 405.72 | 1.90 | ✓ | **22.54** | **0.7756** | **24.60** | **0.7692** | **23.57** | **0.7724** |

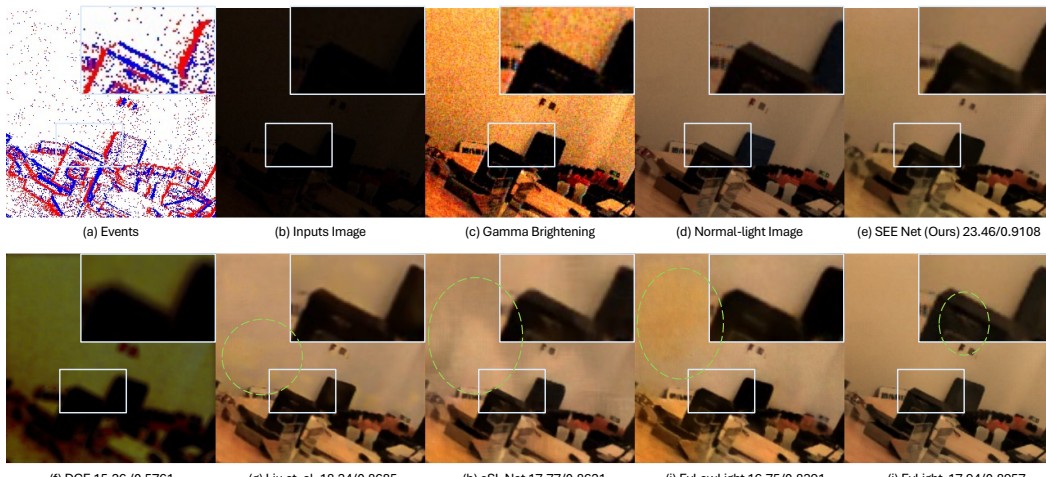

Figure 5: Visualization results on the SDE dataset.

**Dataset:** We conduct experiments on two real-world datasets: **(1) SDE** (Liang et al., 2024) comprises 91 scenes, with 76 for training and 15 for testing. Each scene includes a pair of low-light and normal-light images along with their corresponding events. **(2) SEE-600K** consists of 202 scenes, with each scene containing an average of four sets of videos under different lighting conditions, ranging from low light to bright light. During each training session, we randomly select one set of normal-light images as the reference and use the remaining sets as inputs. For example, for one scene with one low-light, two normal-light, and one high-light set, we generate six pairs of training data.

**Comparative Methods:** We categorize the approaches we compare into four groups. Firstly, DCE (Guo et al., 2020) is a classical approach that can adjust the image brightness curve to achieve normal lighting. Secondly, there are strategies that only use images as input, including SNR (Xu et al., 2022), UFormer (Wang et al., 2022), LLFlow (Wu et al., 2023), and RetinexFormer (Cai et al., 2023). Thirdly, we consider methods that rely solely on events, *e.g.*, E2VID+ (Stoffregen et al., 2020). Tertiary, we examine event-guided low-light enhancement frameworks. This group includes single-frame input methods, *e.g.*, eSL-Net (Wang et al., 2020), Liu et al. (2023), Wang et al. (2023a) and EvLight (Liang et al., 2024), as well as multi-frame input strategies like EvLowLight (Liang et al., 2023). Furthermore, we also compared the HDR reconstruction method HDRevYang et al. (2023). We retrain all methods, following the open-source code when available; for approaches without open-source code, we replicate them based on their respective papers.

**Comparative on SDE Dataset:** The results from our comparative experiments, shown in Tab. 1, reveal several key insights: **(1) performance limitations of single-modal methods**: Methods utilizing only one modality exhibit limited performance, as shown in Tab. 1. This trend underscores the

Table 2: Evaluation on the SEE-600K dataset, with methods trained on both the SDE and SEE-600k. *EvLowLight † refers to this method trained after downsampling the dataset by 10 times.*

| Training Dataset | Methods | low light | | | high light | | | normal light | | |
|---|---|---|---|---|---|---|---|---|---|---|
| | | PSNR | SSIM | $L_1$ | PSNR | SSIM | $L_1$ | PSNR | SSIM | $L_1$ |
| SDE | DCE (Guo et al., 2020) | 9.10 | 0.0968 | 0.3572 | 6.26 | 0.3419 | 0.4649 | **10.79** | **0.3992** | **0.2524** |
| | eSL Net (Wang et al., 2020) | 11.92 | 0.3275 | 0.2703 | **6.66** | **0.1672** | **0.4001** | 7.65 | 0.2685 | 0.3481 |
| | Liu et al. (2023) | 12.41 | 0.4001 | 0.2487 | 5.53 | 0.1950 | 0.4534 | 6.58 | 0.2805 | 0.4129 |
| | EvLowLight (Liang et al., 2023) | 12.68 | 0.4341 | 0.2338 | 4.11 | 0.3071 | 0.6062 | 7.01 | 0.3950 | 0.4520 |
| | EvLight (Liang et al., 2024) | 13.07 | 0.4651 | 0.2337 | 5.12 | 0.1005 | 0.4842 | 6.29 | 0.2805 | 0.4336 |
| | SEENet | **14.84** | **0.5693** | **0.1779** | 3.84 | 0.2119 | 0.6123 | 5.36 | 0.2980 | 0.5056 |
| SEE | eSL Net (Wang et al., 2020) | 11.95 | 0.3845 | 0.2421 | 12.84 | 0.4660 | 0.2076 | 13.45 | 0.5682 | 0.1957 |
| | *EvLowLight † (Liang et al., 2023)* | 12.83 | 0.4511 | 0.2151 | 12.79 | 0.4696 | 0.2084 | 13.04 | 0.5531 | 0.2144 |
| | Liu et al. (2023) | 13.48 | 0.5068 | 0.1946 | 12.30 | 0.4766 | 0.2221 | 13.70 | 0.5474 | 0.2151 |
| | EvLight (Liang et al., 2024) | 13.70 | 0.5150 | 0.1960 | 13.45 | 0.4918 | 0.1990 | 13.63 | 0.5924 | 0.2004 |
| | SEENet | **18.77** | **0.6303** | **0.0971** | **19.21** | **0.6675** | **0.0806** | **20.92** | **0.8002** | **0.0606** |

(a) Events (b) Inputs Image (Low-light) (c) Normal-light Image (d) SEE Net (Ours) 27.17/0.7761

(e) Liu et. al. 15.98/0.6205 (f) eSL Net 14.32/0.5304 (g) EvLight 26.65/0.7145 (h) SEE Net (Ours) Prompt 0.5

(i) Events (j) Inputs Image (Low-light) (k) Normal-light Image (l) SEE Net (Ours) 26.85/0.8937

(m) Liu et. al. 18.90/0.6032 (n) eSL Net 17.30/0.5386 (o) EvLight 16.63/0.4684 (p) SEE Net (Ours) Prompt 0.5

Figure 6: Visual examples of low-light enhancement and high-light recovery on the SEE-600K dataset.

necessity of integrating both modalities for enhanced results, as shown in Fig. 5 (f). **(2) effectiveness of event-guided methods**: In contrast, event-guided image methods demonstrate significantly better performance. These approaches leverage the complementary strengths of both events and traditional images, leading to better outcomes in low-light conditions, as shown in Fig. 5 (g-j). **(3) impact of indoor and outdoor conditions**: Notably, performance in low-light indoor scenarios is inferior to that in outdoor settings, as shown in Fig. 5 (e). This discrepancy may be attributed to the issues of flickering light sources commonly found indoors (Xu et al., 2023a). Our SEE-Net consistently achieves the best results across both scenarios, with a model size of just $1.9M$—10% parameter count of other SOTA methods—demonstrating its efficiency and compactness in low-light image enhancement.

**Comparative on SEE-600K Dataset:** The results presented in Tab. 2 illustrate the performance of various methods across different lighting conditions on the SEE-600K dataset. **(1) trained on SDE**: Models trained on the SDE dataset maintain a reasonable level of performance when tested on the SEE-600K dataset, particularly in low-light conditions. Notably, the DCE Guo et al. (2020) achieves the best results in high-light scenarios, underscoring its excellent generalization capabilities for its self-supervised approach. **(2) trained on SEE-600K**: Models trained on the SEE-600K dataset exhibit improved performance in both low-light and high-light conditions. Our proposed SEE-Net method stands out as the best performer, as shown in Tab.2 and Fig. 6. This achievement is due to our innovative use of prompt adjustments, which effectively resolve the ambiguity often seen in

Table 3: Ablation studies. $f_c$ indicates cross-attention. $f_{pe}$ stands for learning-based embedding.

| Case | Bayer Pattern | Encoding | Loop | Prompt Embedding | Cascade | Prompt Merge | PSNR | SSIM |
|------|------|------|------|------|------|------|------|------|
| 1 | $f_{pe}$ | $f_{ca}$ | 20 | $f_{pe}$ | ✓ | + | 23.57 | 0.7724 |
| 2 | - | $f_{ca}$ | 20 | $f_{pe}$ | ✓ | + | 22.94 | 0.7686 |
| 3 | $f_{pe}$ | $add + conv$ | 20 | $f_{pe}$ | ✓ | + | 22.40 | 0.7224 |
| 4 | $f_{pe}$ | $cat + conv$ | 20 | $f_{pe}$ | ✓ | + | 22.84 | 0.7298 |
| 5 | $f_{pe}$ | $f_{ca}$ | 10 | $f_{pe}$ | ✓ | + | 22.18 | 0.6812 |
| 6 | $f_{pe}$ | $f_{ca}$ | 20 | $sin$ | ✓ | + | 23.08 | 0.7692 |
| 7 | $f_{pe}$ | $f_{ca}$ | 20 | $f_{pe}$ | ✗ | + | 22.26 | 0.7713 |
| 8 | $f_{pe}$ | $f_{ca}$ | 20 | $f_{pe}$ | ✓ | × | 22.94 | 0.7893 |

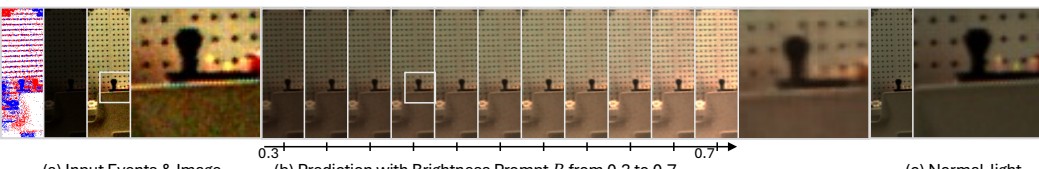

(a) Input Events & Image    (b) Prediction with Brightness Prompt $B$ from 0.3 to 0.7    (c) Normal-light

Figure 7: Visualization of brightness adjustment using varying brightness prompts $B$ from 0.3 to 0.7, showing smooth brightness transitions in SEE-600K dataset. For more visualizations, see the Appendix.

enhancement processes. Overall, these results highlight the effectiveness of our approach across diverse lighting conditions, further validating its robustness. **(3) advantages of prompt adjustments:** Unlike previous methods, Fig. 6, that are limited to one-way mapping, our approach with prompt adjustments demonstrates significant advantages, as shown in Fig. 6 (h,p). Prompt adjustments allow us to produce image quality that surpasses the ground truth, Fig. 6 (d,i), regardless of whether low-light or high-light conditions are used as input. When the prompt is set to 0.5, the output achieves optimal brightness and sharp textures. *For additional visualization, please refer to the appendix.*

**Ablation and Analytical Studies:** In this ablation study (Tab.3), we analyze the impact of various components using Case #1 as the baseline. **(1) bayer pattern embedding**: Removing the bayer-pattern embedding (Case #2) leads to a performance drop, indicating it enhances accuracy but is not the most critical factor. (2) encoding: Replacing the cross-attention module $f_c$ with a convolutional layer in both Case #3 (add) and Case #4 (concat) leads to significant performance degradation, underscoring the critical role of cross-attention. **(3) loop iterations**: Reducing loop iterations from 20 to 10 (Case #4) causes a performance decline, indicating sufficient iterations are necessary for refinement. **(4) prompt embedding**: Switching the prompt embedding from $f_{pe}$ to a sine function (Vaswani, 2017) (Case #5) yields similar performance but doesn't surpass the learned embedding. **(5) prompt merge**: Disabling prompt merge (Case #6) results in a slight performance drop, indicating its importance for optimal results. **(6) multi-prompt adjustment**: Fig.7 shows the output under multiple prompts. The input consists of a low-light image and events. When using gamma correction to brighten the low-light image, significant noise is introduced (Fig.7 (a)). However, our outputs with varying prompts effectively control brightness while reducing noise (Fig.7 (b)), demonstrating the flexibility and robustness of our method in post-processing. *Due to space limitations, please refer to the appendix for more information.*

# 7 CONCLUSION

In this paper, we proposed a new research problem: how to use events to adjust the brightness of images across a wide range of lighting conditions, from low light to high light. To address this challenge, we made the following contributions. **(1)**, we developed a physical model and formally defined the problem of brightness adjustment using events, providing a solid theoretical foundation. **(2)**, we introduced a new spatiotemporal registration algorithm based on a robotic arm and collected a large-scale dataset, **SEE-600K**, to overcome alignment issues and support our research. **(3)**, we presented **SEE-Net**, a novel and compact framework capable of accepting input images with a wide range of illumination and producing output images with adjustable brightness. **(4)**, we conducted extensive experiments to demonstrate the effectiveness of our method.

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

Table 4: DVS346 Event Output Specifications (iniVation AG, 2021)

| Parameter | Value |
|---|---|
| Spatial resolution | $346 \times 260$ pixels |
| Temporal resolution | $1\,\mu$s |
| Maximum throughput | 12 million events per second (MEPS) |
| Typical latency | $< 1\,$ms |
| Dynamic range | Approx. $120\,$dB |
| Contrast sensitivity | $14.3\%$ (ON events), $22.5\%$ (OFF events) |

# Appendix

To address the reviewers' feedback, we have added the following three sections to the supplementary material:

**(1) More Details About the DVS346 Sensor:** We provide additional information on the sensor's specifications, particularly regarding noise and image quality, to contextualize the limitations of the APS frames in our dataset.

**(2) Differences Between Brightness Adjustment and HDR Reconstruction:** We clarify the differences between our brightness adjustment task and HDR reconstruction, focusing on objectives, challenges, and data construction methods, supported by mathematical formulations.

**(3) Output Visualizations of Different Prompts:** We include visual examples showing how our network processes inputs under extreme low-light and high-light conditions using various brightness prompts, directly addressing how the brightness prompt $B$ influences the outputs from different input images.

In the final paper, we will organize the Appendix accordingly. For now, we have placed these sections at the beginning of the supplementary material for the reviewers' convenience.

## A    MORE DETAILS ABOUT THE DVS346 SENSOR

In our experiments, we employed the DVS346 event camera, a sensor capable of simultaneously outputting asynchronous events and synchronous image frames (APS frames). Despite its widespread use in the academic community, the DVS346 has inherent limitations that affect the quality of the captured images, particularly due to various noise factors. Understanding these parameters is crucial for contextualizing the performance of our proposed methods.

The specifications of the DVS346 sensor are detailed in Tables 4 and 5. Below, we explain the significance of each parameter, emphasizing those related to noise, to illustrate the image quality from this sensor.

**Events:** *Spatial resolution:* refers to the number of pixels in the sensor array, which in this case is $346 \times 260$ pixels. *Temporal resolution:* of $1\,\mu$s indicates the sensor's ability to detect rapid changes in brightness, allowing for precise temporal event detection. This high temporal resolution is advantageous for capturing fast-moving scenes. *Maximum throughput:* of 12 MEPS means the sensor can handle up to 12 million events per second, which is essential for recording scenes with a lot of motion without losing data. *Typical latency:* of less than $1\,$ms ensures minimal delay between the occurrence of an event and its registration by the sensor, which is important for real-time applications. *Dynamic range:* of approximately $120\,$dB allows the event sensor to operate effectively under a wide range of lighting conditions, from very dark to very bright environments. This high dynamic range is a key advantage of event-based cameras. *Contrast sensitivity:* represents the minimum percentage change in brightness required to generate an event. The sensor has a contrast sensitivity of $14.3\%$ for ON events and $22.5\%$ for OFF events. While higher contrast sensitivity reduces noise by preventing the sensor from triggering on minor fluctuations, it may also cause it to miss subtle changes in brightness.

**Frame:** *Spatial resolution:* for the APS frames is the same as the event output, limiting the detail in the captured images. **Frame rate:** of $40\,$FPS indicates that the sensor captures 40 frames per second.

Table 5: DVS346 Frame Output Specifications (iniVation AG, 2021)

| Parameter | Value |
|---|---|
| Spatial resolution | $346 \times 260$ pixels |
| Frame rate | $40, \text{FPS}$ |
| Dynamic range | $55, \text{dB}$ |
| Fixed Pattern Noise (FPN) | $4.2\%$ |
| Dark signal | $18{,}000, \text{e}^-/\text{s}$ |
| Readout noise | $55, \text{e}^-$ |

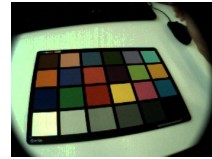 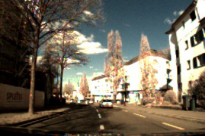 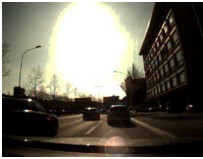 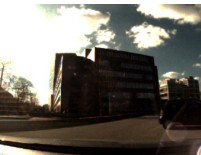 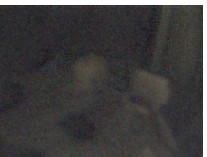

| Image saturation deviation | Color noise on trees | Overexposure of sky | Underexposure of buildings | Low-light noise |

Figure 8: Examples of other datasets from the event-based vision community (Scheerlinck et al., 2019; Cui et al., 2024; Wang et al., 2023b; Liang et al., 2024). Although the DVS 346 camera suffers from insufficient dynamic range and noise, it is still the data acquisition device that can best support the training of various event vision tasks at this stage.

***Dynamic range:*** of $55, \text{dB}$ is significantly lower than that of the event output. This limited dynamic range means the APS frames struggle with scenes that have both very bright and very dark areas, leading to overexposure or underexposure in parts of the image. ***Fixed Pattern Noise (FPN):*** of $4.2\%$ refers to the non-uniformity in pixel responses, where each pixel may have a slightly different baseline level of response due to manufacturing inconsistencies. High FPN manifests as a static noise pattern over the image, degrading the visual quality. ***Dark signal:*** of $18{,}000, \text{e}^-/\text{s}$ represents the amount of charge accumulated by a pixel in the absence of light. A high dark signal increases the baseline noise level, especially noticeable in low-light conditions, resulting in grainy images. ***Readout noise:*** of $55, \text{e}^-$ is the noise introduced during the process of reading the pixel values from the sensor. This electronic noise adds uncertainty to the pixel values, further reducing image clarity and detail, particularly in darker regions where the signal level is low.

**Impact on Image Quality:** The combination of these parameters adversely affects the image quality of the APS frames produced by the DVS346 sensor: (1) A **dynamic range** of $55$ dB is insufficient for high-contrast scenes, causing loss of detail in shadows (underexposure) or highlights (overexposure). This limitation means that the APS frames cannot effectively capture scenes with both bright and dark regions simultaneously. (2) High levels of **Fixed Pattern Noise** introduce consistent noise patterns across the image, which are difficult to remove and can be distracting in the final output. (3) The significant **dark signal** contributes to increased noise, especially in low-light conditions where the actual signal from the scene is weak. This results in a lower signal-to-noise ratio (SNR), making the images appear grainy or speckled. (4) Elevated **readout noise** further degrades image quality by adding random variations to the pixel values during the readout process, obscuring fine details and reducing overall sharpness.

These noise-related issues collectively lead to suboptimal image quality in the APS frames, with noticeable artifacts such as blurriness, graininess, and loss of detail. Understanding the limitations of the DVS346 sensor is essential for interpreting the results of our research. While the sensor's APS frames have quality constraints due to noise and limited dynamic range, the event output excels in capturing high temporal resolution and wide dynamic range changes. Our work leverages the strengths of the event data to adjust image brightness across various lighting conditions, mitigating some of the APS frame limitations.

Despite the challenges posed by the sensor's noise characteristics, the DVS346 remains a valuable tool in event-based vision research (Scheerlinck et al., 2019; Cui et al., 2024; Wang et al., 2023b; Liang

et al., 2024) due to its accessibility and the richness of the event data it provides, as shown in Fig. 8. As technology advances, we anticipate that future sensors will offer improved image quality with reduced noise levels, enhancing the potential for high-quality event-based imaging. In the meantime, acknowledging and addressing these limitations allows us to develop algorithms that compensate for the sensor's shortcomings, contributing to the advancement of event-based vision applications.

## B DIFFERENCES BETWEEN BRIGHTNESS ADJUSTMENT AND HDR RECONSTRUCTION

In this section, we discuss the fundamental differences between our proposed brightness adjustment task using event cameras and the traditional High Dynamic Range (HDR) reconstruction task. We highlight the distinctions in objectives, challenges, and data construction methodologies, supported by mathematical formulations for clarity.

**Different Objectives:**

The primary goal of HDR reconstruction is to expand the dynamic range of an image, capturing details in both dark and bright regions that exceed the capability of standard Low Dynamic Range (LDR) sensors. Mathematically, HDR imaging seeks to recover a radiance map $R(x)$ that represents the true scene radiance over a wide dynamic range:

$$R(x) = f^{-1}(I_{\text{LDR}}(x)), \tag{7}$$

where $I_{\text{LDR}}(x)$ is the observed LDR image, and $f^{-1}$ is the inverse of the camera response function.

In contrast, our brightness adjustment task focuses on modifying the exposure level of an image to enhance visibility and recover lost details due to underexposure or overexposure, without necessarily expanding the dynamic range. The objective is to obtain an adjusted image $\hat{I_{\text{rgb}}}$ from an input image $I_{rgb}$ and event data $E(x, t)$:

$$\hat{I_{\text{rgb}}} = f_{see}(I_{rgb}, E; B), \tag{8}$$

where $f_{see}$ is our proposed adjustment function, $E$ represents the event stream, and $B$ is the brightness prompt controlling the desired exposure level.

**Different Challenges:**

HDR reconstruction faces the challenge of accurately merging multiple images captured at different exposure levels to create a single image with an expanded dynamic range. This often requires precise alignment and handling of motion between exposures to avoid ghosting artifacts. The mathematical formulation involves combining $N$ images $\{I_i(x)\}_{i=1}^{N}$ with corresponding exposure times $\{t_i\}_{i=1}^{N}$:

$$R(x) = \frac{\sum_{i=1}^{N} w(I_i(x)) \cdot f^{-1}(I_i(x))}{\sum_{i=1}^{N} w(I_i(x))}, \tag{9}$$

where $w(I_i(x))$ is a weighting function that emphasizes well-exposed pixels.

Our brightness adjustment task, on the other hand, deals with the challenge of adjusting images captured under various lighting conditions using the high temporal resolution and dynamic range of event data. Unlike HDR reconstruction, we do not require multiple images at different exposures. Instead, we leverage events to infer illumination changes and guide the brightness adjustment of a single input image. The adjustment function $f_{see}$ must effectively fuse spatial image data and temporal event information:

$$I_{\text{adj}}(x) = f_d\left(f_{se}(I_{rgb}, E), B\right), \tag{10}$$

where $f_{se}$ is an encoder that extracts features from the input image and events, and $f_d$ is a decoder that generates the adjusted image based on the brightness prompt $B$.

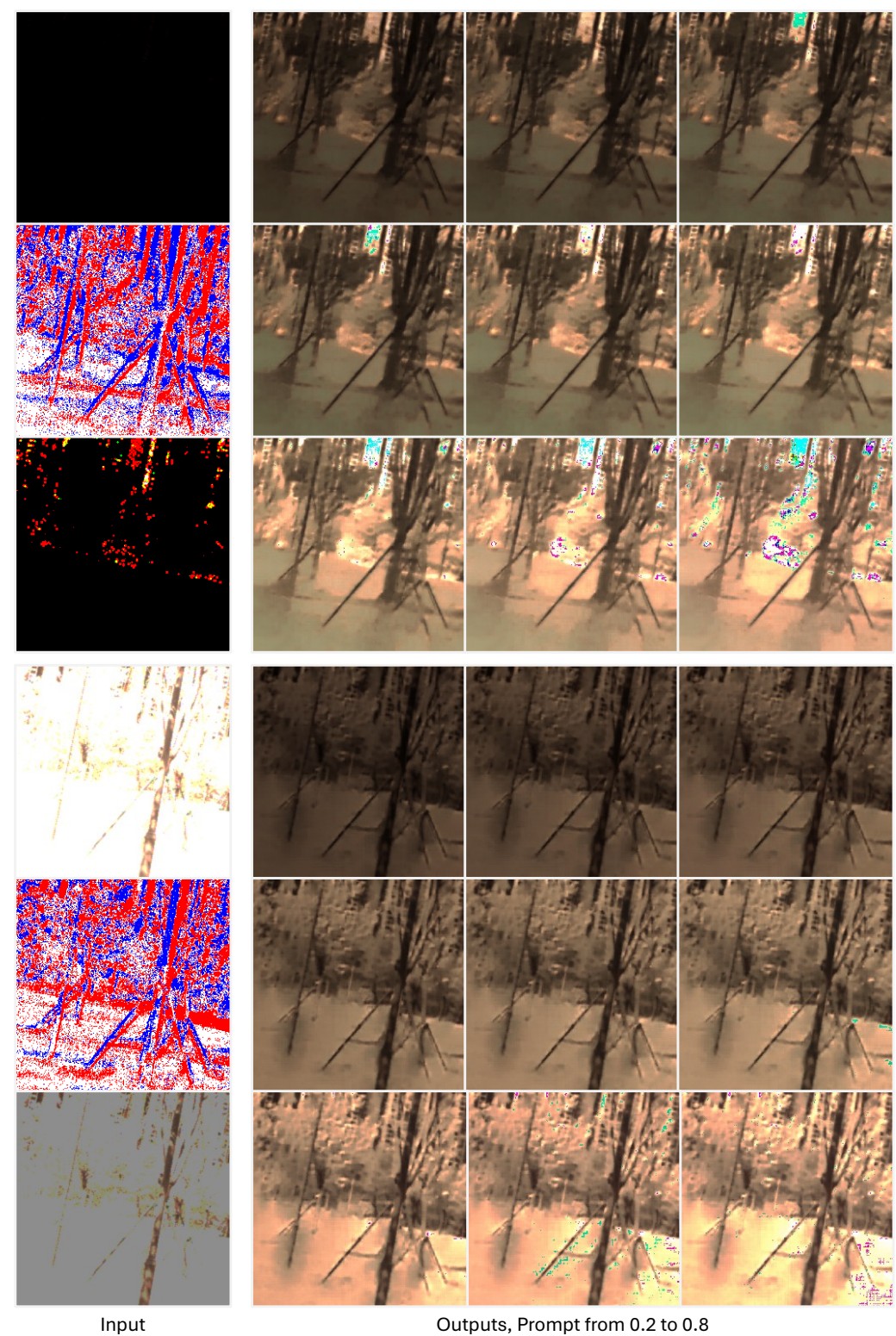

Input          Outputs, Prompt from 0.2 to 0.8

Figure 9: Under the same scene, with low-light and high-light images as inputs, we compare the outputs generated using a series of prompts. The inputs are the original image, events, and the visualization of the original image after gamma correction. Almost all the contours and details in the original image are lost.

**Different Data Construction Methods:**

Constructing datasets for HDR reconstruction typically involves capturing multiple images of the same scene at different exposure levels, requiring static scenes or sophisticated alignment techniques to handle motion. The ground truth HDR image is often synthesized by merging these exposures.

Mathematically, for each scene, we collect $N$ images:

$$\{I_i(x)\}_{i=1}^N, \quad \text{with exposure times } t_1 < t_2 < \cdots < t_N, \tag{11}$$

and compute the ground truth radiance $R(x)$ as shown earlier.

For our brightness adjustment task, data construction is more straightforward and scalable. We capture pairs of images and corresponding event data under varying lighting conditions using Neutral Density (ND) filters to simulate different exposures. Each scene provides synchronized data without the need for multiple exposure times or complex alignment:

$$(I_{rgb}, E, \hat{I_{\text{rgb}}}), \tag{12}$$

where $\hat{I_{\text{rgb}}}$ is the ground truth image at the desired exposure. The use of events allows us to handle dynamic scenes effectively, as the high temporal resolution of events captures rapid changes in illumination.

In essence, while HDR reconstruction aims to create images with an expanded dynamic range by combining multiple exposures, our brightness adjustment task seeks to adjust the exposure of images using event data to recover lost details without extending the dynamic range. **Our approach is more practical for real-world applications where capturing multiple exposures is impractical or impossible.**

By formulating the problem differently and leveraging the unique properties of event cameras, we address challenges specific to brightness adjustment under diverse lighting conditions. This includes handling dynamic scenes and providing fine-grained control over image brightness through prompts.

Our dataset construction method is also more scalable, enabling us to create a large dataset without the complexities involved in HDR dataset creation. This allows for training more robust models suited to real-world scenarios.

## C  OUTPUT VISUALIZATIONS OF DIFFERENT PROMPTS

The Fig. 9 demonstrates the effectiveness of our network in reconstructing images across a broad range of lighting conditions using events. We input both extremely low-light and overexposed images—where the original contours and details are significantly degraded or lost—into our network to observe how it handles varying input brightness levels when the same brightness prompt is applied.

Our network leverages the high dynamic range and temporal resolution of events to recover lost details in both underexposed and overexposed scenarios. By integrating events, which captures pixel-level changes in brightness over time, the network compensates for the deficiencies of the input images regardless of their initial exposure levels.

We present the results corresponding to brightness prompts ranging from 0.2 to 0.8, allowing for fine-grained control over the brightness of the output images. Each prompt value is applied to both the extremely low-light and overexposed input images. Despite the drastic differences in the original brightness of the inputs, the outputs generated with the same brightness prompt are remarkably consistent in terms of exposure and detail.

This observation directly answers the reviewer's question: when reconstructing a bright image (*e.g.*, setting $B = 0.8$) from two different input images—one dark and one bright—the network produces output images that are both well-exposed and visually similar. Although the low light input image produced some artifacts. This demonstrates that the output is primarily determined by the brightness prompt $B$, rather than the original brightness of the input images. The network effectively adjusts the input images to the desired brightness level specified by the prompt, utilizing the event data to recover or suppress details as needed.

The output of Fig. 9 includes nine groups of results, each corresponding to a different brightness prompt. Overall, the figure underscores the robustness and flexibility of our network. It highlights the capability to use event data effectively for restoring details lost in extreme lighting conditions while providing precise brightness control through prompts. This adaptability makes our approach highly suitable for applications requiring image enhancement across diverse lighting environments, ensuring consistent output quality regardless of the initial exposure of the input images.

# D   MORE DETAILS FOR RESEARCH PROBLEM DEFINITION

Imaging is the process of capturing light from a scene, which can be represented as a radiance field $L(t)$ that varies over time $t$. The intensity of ambient light in real-world environments spans a wide range, from approximately $0.1$ lux in low-light conditions to over $1e6$ lux under bright sunlight. The goal of our learning task is to accurately recover $L(t)$ and transform it into a visual representation that is suitable for human perception.

**Sensor Signal Acquisition and Noise Modeling:**

Cameras equipped with active pixel sensors record light signals through an exposure process. During the exposure time $t_e$, the sensor integrates incoming photons to produce a voltage $V$. The number of photons $k$ detected is a random variable following a Poisson distribution due to the quantum nature of light:

$$k \sim \mathcal{P}(\lambda), \quad \lambda = \eta \int_{t_e} L(t)dt, \tag{13}$$

where:

- $\lambda$ is the expected number of photons,
- $\eta$ is the quantum efficiency of the sensor,
- $L$ is the light intensity,
- $t_e$ is the exposure time.

The voltage $V$ generated by the sensor is proportional to the number of detected photons and is given by:

$$V = Gk + N_d, \tag{14}$$

where:

- $G$ is the sensor gain, usually a circuit amplifier,
- $N_d \sim \mathcal{N}(\mu_d, \sigma_d^2)$ represents the dark current noise, typically modeled as Gaussian noise with mean $\mu_d$ and variance $\sigma_d^2$.

The RAW image intensity $I_{\text{raw}}$ is obtained by quantize the voltage $V$:

$$I_{\text{raw}} = \mathcal{Q}(V) = \mathcal{Q}(Gk + N_d), \tag{15}$$

where $\mathcal{Q}$ is the quantization function converting continuous voltage signals into discrete digital values, typically ranging from 8 *bits* to 14 *bits*.

**Image Signal Processing (ISP)**

The RAW image $I_{\text{raw}}$ undergoes an image signal processing pipeline $f_{\text{isp}}$ that includes steps such as denoising (Buades et al., 2005), demosaicing (Li et al., 2008), color correction (Gasparini & Schettini, 2003), and tone mapping (Debevec & Gibson, 2002) to produce the final RGB image:

$$I_{\text{rgb}} = f_{\text{isp}}(I_{\text{raw}}). \tag{16}$$

**Characteristics of Accurate Exposure**

An accurate exposure process aims to produce $I_{\text{rgb}}$ with the following characteristics:

1. **Accurate Exposure**: The mean pixel intensity of $I_{\text{rgb}}$ falls within a desirable range for human observation, typically normalized between 0.4 and 0.7 (Mertens et al., 2009):

$$0.4 \le \frac{1}{N} \sum_{i=1}^{N} I_{\text{rgb}}^{(i)} \le 0.7, \tag{17}$$

where $N$ is the total number of pixels.

2. **Noise-Free**: The influences of dark current noise $N_d$ and photon shot noise $N_s$ are minimized or eliminated:

$$\text{Var}(I_{\text{rgb}}) \approx \text{Var}(G\eta \int_{t_e} L(t)dt), \tag{18}$$

implying that the variance due to noise is negligible.

3. **Color Neutrality**: The image has no color cast; the grayscale values computed from each RGB channel are approximately equal (Buchsbaum, 1980):

$$f_{\text{gray}}(I_r) \approx f_{\text{gray}}(I_g) \approx f_{\text{gray}}(I_b), \tag{19}$$

where $I_r$, $I_g$, and $I_b$ are the red, green, and blue channels of $I_{\text{rgb}}$, and $f_{\text{gray}}$ is a function mapping RGB values to grayscale.

**Limitations of Traditional Cameras**

Traditional cameras have a limited dynamic range of approximately $80dB$, which often results in loss of detail in scenes with high contrast. Under extreme lighting conditions, images may exhibit overexposed highlights or underexposed shadows, leading to insufficient edge and texture information.

**Advantages of Event Cameras**

Event cameras overcome these limitations by offering:

- **High Dynamic Range**: Greater than $120\,\text{dB}$, allowing them to handle extreme lighting variations.
- **High Temporal Resolution**: Less than 1 ms, enabling them to capture fast-changing scenes.

Event cameras operate asynchronously by detecting changes in illumination at each pixel. The output is a stream of events, each represented as:

$$(x, y, t, p), \tag{20}$$

where:

- $(x, y)$ are the pixel coordinates,
- $t$ is the timestamp,
- $p \in \{+1, -1\}$ indicates the polarity (increase or decrease in light intensity).

**Event Generation Mechanism**

An event is generated at a pixel $(x, y)$ when the change in the logarithm of the light intensity exceeds a predefined threshold $C$:

$$\Delta L(x, y, t) = \log(L(x, y, t)) - \log(L(x, y, t_k)) = pC, \tag{21}$$

where:

- $L(x, y, t)$ is the light intensity at time $t$,
- $t_k$ is the timestamp of the last event at pixel $(x, y)$,
- $p$ is the polarity,
- $C$ is the contrast sensitivity threshold.

This condition can also be expressed in terms of relative intensity change:

$$\frac{L(x, y, t)}{L(x, y, t_k)} = e^{pC}. \tag{22}$$

**Proposed Model for Illumination Recovery**

Given the high dynamic range and temporal resolution of event cameras, we aim to utilize an images $I_{\text{rgb}}$ and corresponding events $E$ to recover the scene's illumination $L(t)$ and present it in a human-friendly format. However, due to the extensive theoretical range of $L(t)$, we introduce a brightness control prompt $B$ to adjust the output image's mean brightness.

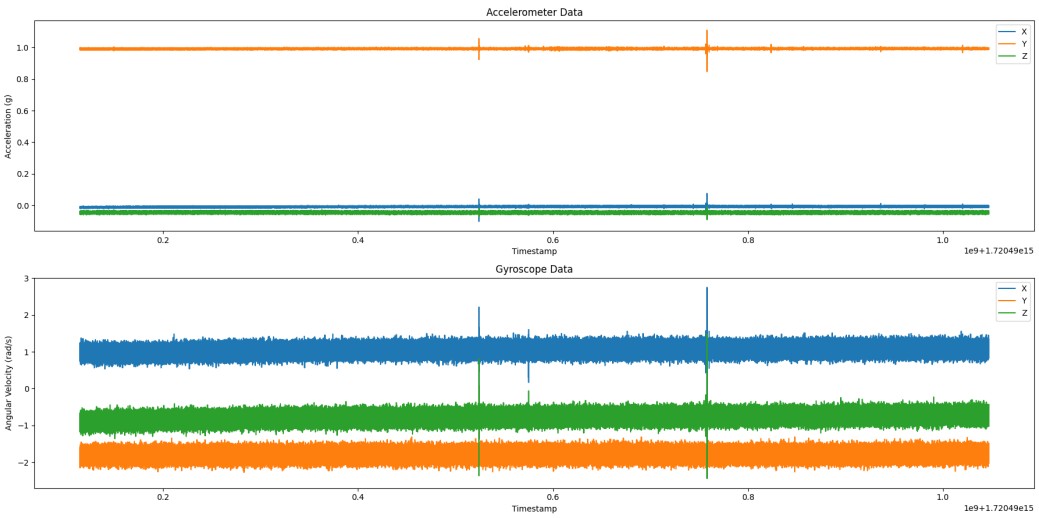

Figure 10: The IMU sensor is calibrated by leaving the sensor alone for about one hour to obtain the deviations of the IMU in various directions.

Our model is defined as:

$$\hat{I_{\text{rgb}}} = f_{\text{see}}(I_{\text{rgb}}, E, B),\qquad(23)$$

where:

- $f_{\text{see}}$ is a function designed to enhance the input image $I_{\text{rgb}}$ using the events $E$ and adjust the brightness according to $B$,

- $\hat{I}_{\text{rgb}}$ is the output image with improved exposure,

- $B$ is a user-defined parameter representing the desired mean brightness of $\hat{I}_{\text{rgb}}$:

$$B = \frac{1}{N}\sum_{i=1}^{N}I_{\text{rgb}}^{\hat{(i)}}\qquad(24)$$

**Benefits of the Proposed Approach**

1. **Robust Training**: By presetting the parameter $B$ during the training phase, the model can mitigate biases present in the training dataset, leading to more generalized performance.

2. **Flexibility in Usage**: During inference, setting $B = 0.5$ (assuming pixel values are normalized between 0 and 1) aligns with common exposure levels, but users can adjust $B$ for creative control over the image's brightness and exposure, enabling image adjustments and editing capabilities.

## E    TEMPORAL IMU REGISTRATION ALGORITHM

In this section, we provide a more detailed description of our IMU data registration algorithm, which aligns a source sequence $S$ and a target sequence $T$ by finding the optimal temporal bias $b$ and matching length $l$ that minimize the distance between them. Due to the high sampling rate of IMU data (1000 $Hz$), an exhaustive search over all possible biases is computationally prohibitive. Therefore, we introduce a multi-level iterative strategy that efficiently approximates the optimal alignment.

**IMU Data Calibration and Stability**

Fig. 10 illustrates the calibration results of our IMU sensor over a one-hour period during which the sensor remained stationary. From this figure, we observe that the IMU's measurement errors are stable over long durations and do not increase over time. The deviations in the accelerometer's three axes and the gyroscope's three axes are consistent, indicating reliable sensor performance. Through calibration,

Table 6: Calibration results showing biases, variances, and standard deviations for each axis of the accelerometer and gyroscope.

| Sensor | Axis | Bias | Variance | Standard Deviation |
|--------|------|------|----------|--------------------|
| Accelerometer | X | $-0.009256$ | $5.836 \times 10^{-6}$ | $0.002416$ |
| Accelerometer | Y | $0.993344$ | $6.196 \times 10^{-6}$ | $0.002489$ |
| Accelerometer | Z | $-0.048622$ | $1.348 \times 10^{-5}$ | $0.003672$ |
| Gyroscope | X | $1.081781$ | $0.010550$ | $0.102711$ |
| Gyroscope | Y | $-1.791223$ | $0.011102$ | $0.105365$ |
| Gyroscope | Z | $-0.697237$ | $0.011360$ | $0.106582$ |

we corrected these biases during preprocessing to enhance measurement accuracy. Specifically, for the camera used in our dataset collection, the calibrated IMU errors are quantified shown in Tab. 6. These low variance values indicate that the IMU's measurement noise is within an acceptable and small range, affirming that our calibration process effectively corrects sensor deviations. Consequently, we can achieve accurate results in our data registration by leveraging the stability of the IMU sensor. The specific implementation steps of our calibration process are detailed below.

**IMU Data Preprocessing with Kalman Filter**

We first denoise the raw IMU data using a Kalman filter Mirzaei & Roumeliotis (2008). For each IMU sequence (source and target), we model the system as:

$$\mathbf{x}_k = \mathbf{F}\mathbf{x}_{k-1} + \mathbf{w}_{k-1}, \tag{25}$$

$$\mathbf{z}_k = \mathbf{H}\mathbf{x}_k + \mathbf{v}_k, \tag{26}$$

where $\mathbf{x}_k \in \mathbb{R}^6$ is the state vector at time $k$, consisting of accelerometer and gyroscope measurements:

$$\mathbf{x}_k = \begin{bmatrix} \mathrm{acc}_x \\ \mathrm{acc}_y \\ \mathrm{acc}_z \\ \mathrm{gyr}_x \\ \mathrm{gyr}_y \\ \mathrm{gyr}_z \end{bmatrix}_k,$$

$\mathbf{F} \in \mathbb{R}^{6 \times 6}$ is the state transition matrix (identity matrix in our case), $\mathbf{w}_{k-1}$ is the process noise with covariance $\mathbf{Q}$, $\mathbf{z}_k \in \mathbb{R}^6$ is the measurement vector, $\mathbf{H}$ is the observation matrix (also identity), and $\mathbf{v}_k$ is the measurement noise with covariance $\mathbf{R}$.

The Kalman filter recursively estimates the state $\mathbf{x}_k$ by:

$$\text{Prediction Step:} \quad \hat{\mathbf{x}}_{k|k-1} = \mathbf{F}\hat{\mathbf{x}}_{k-1|k-1}, \tag{27}$$

$$\mathbf{P}_{k|k-1} = \mathbf{F}\mathbf{P}_{k-1|k-1}\mathbf{F}^\top + \mathbf{Q}, \tag{28}$$

$$\text{Update Step:} \quad \mathbf{K}_k = \mathbf{P}_{k|k-1}\mathbf{H}^\top(\mathbf{H}\mathbf{P}_{k|k-1}\mathbf{H}^\top + \mathbf{R})^{-1}, \tag{29}$$

$$\hat{\mathbf{x}}_{k|k} = \hat{\mathbf{x}}_{k|k-1} + \mathbf{K}_k(\mathbf{z}_k - \mathbf{H}\hat{\mathbf{x}}_{k|k-1}), \tag{30}$$

$$\mathbf{P}_{k|k} = (\mathbf{I} - \mathbf{K}_k\mathbf{H})\mathbf{P}_{k|k-1}, \tag{31}$$

where $\hat{\mathbf{x}}_{k|k}$ is the estimated state at time $k$, $\mathbf{P}_{k|k}$ is the estimated covariance, and $\mathbf{K}_k$ is the Kalman gain.

The initial state $\hat{\mathbf{x}}_{0|0}$ is set to the first measurement, and the initial covariance $\mathbf{P}_{0|0}$ is set to the identity matrix.

**Multi-Level Downsampling**

To reduce computational complexity, we create two additional levels of downsampled sequences using average pooling:

- Level-1: Downsampled by a factor of $s_1$.
- Level-2: Downsampled by a factor of $s_1 \times s_2$.

The downsampling is performed by averaging over non-overlapping windows of size $s_i$, for $i = 1, 2$. For example, for Level-1, the downsampled sequence $S_1$ is obtained as:

$$S_1[n] = \frac{1}{s_1} \sum_{k=(n-1)s_1+1}^{ns_1} S[k], \quad n = 1, 2, \ldots, \left\lfloor \frac{L_S}{s_1} \right\rfloor, \tag{32}$$

where $L_S$ is the length of the original sequence $S$.

**Hierarchical Bias Search**

At each level, we perform a search for the optimal temporal bias $b$ and matching length $l$ that minimize the distance between the source and target sequences.

*Distance Metric*

We define the distance between two sequences $S$ and $T$ over a matching window of length $l$ as the mean Euclidean distance between their accelerometer and gyroscope data:

$$d_{\mathrm{acc}}(S, T; b, l) = \frac{1}{l} \sum_{k=1}^{l} \left\| \mathbf{a}_S[k + b] - \mathbf{a}_T[k] \right\|_2, \tag{33}$$

$$d_{\mathrm{gyr}}(S, T; b, l) = \frac{1}{l} \sum_{k=1}^{l} \left\| \mathbf{g}_S[k + b] - \mathbf{g}_T[k] \right\|_2, \tag{34}$$

where $\mathbf{a}_S[k]$ and $\mathbf{g}_S[k]$ are the accelerometer and gyroscope measurements of sequence $S$ at time $k$, respectively.

*Coarse Search at Level-2*

At the lowest resolution (Level-2), we perform a coarse search over a large range of biases $b$:

$$b \in [b_{\min}, b_{\max}], \tag{35}$$

where $b_{\min}$ and $b_{\max}$ are chosen based on the expected maximum temporal misalignment.

For each candidate bias $b$, we compute the distances $d_{\mathrm{acc}}$ and $d_{\mathrm{gyr}}$ and record the bias that minimizes these distances:

$$b_{\mathrm{acc}}^{(2)} = \arg\min_b d_{\mathrm{acc}}(S_2, T_2; b, l_b), \tag{36}$$

$$b_{\mathrm{gyr}}^{(2)} = \arg\min_b d_{\mathrm{gyr}}(S_2, T_2; b, l_b), \tag{37}$$

where $l_b$ is the matching length at bias $b$, determined by the overlapping length of the sequences after applying the bias.

*Refined Search at Level-1 and Level-0*

Using the biases obtained at Level-2 as center points, we perform refined searches at higher resolutions (Level-1 and Level-0). The search ranges at each higher level are narrowed down around the biases found at the previous level:

$$b_{\min}^{(i)} = b^{(i+1)} - \delta^{(i)}, \tag{38}$$

$$b_{\max}^{(i)} = b^{(i+1)} + \delta^{(i)}, \quad i = 1, 0, \tag{39}$$

where $\delta^{(i)}$ is a small range that depends on the downsampling factor.

At each level, we update the biases:

$$b_{\text{acc}}^{(i)} = \arg \min_{b \in [b_{\text{min}}^{(i)}, b_{\text{max}}^{(i)}]} d_{\text{acc}}(S_i, T_i; b, l_b), \tag{40}$$

$$b_{\text{gyr}}^{(i)} = \arg \min_{b \in [b_{\text{min}}^{(i)}, b_{\text{max}}^{(i)}]} d_{\text{gyr}}(S_i, T_i; b, l_b), \tag{41}$$

for $i = 1, 0$.

**Optimal Bias and Alignment**

After performing the refined searches, we obtain the optimal biases $b_{\text{acc}}^{(0)}$ and $b_{\text{gyr}}^{(0)}$ at the original data level (Level-0). We choose the final bias $b^*$ and matching length $l^*$ based on the minimum distances:

$$b^* = \text{median}(b_{\text{acc}}^{(0)}, b_{\text{gyr}}^{(0)}), \tag{42}$$

$$l^* = \min(L_S - b^*, L_T), \tag{43}$$

where $L_S$ and $L_T$ are the lengths of the source and target sequences, respectively.

The source and target sequences are then aligned by shifting the source sequence by $b^*$ and taking the first $l^*$ samples:

$$S_{\text{aligned}}[k] = S[k + b^*], \quad k = 1, 2, \ldots, l^*; \tag{44}$$

$$T_{\text{aligned}}[k] = T[k], \quad k = 1, 2, \ldots, l^*. \tag{45}$$

**Algorithm Summary**

The overall algorithm can be summarized as follows:

1. Apply Kalman filter to denoise the source and target IMU sequences.

2. Downsample the sequences to create Level-1 and Level-2 versions.

3. At Level-2, perform a coarse search over a wide range of biases to find initial estimates $b_{\text{acc}}^{(2)}$ and $b_{\text{gyr}}^{(2)}$.

4. At Level-1, perform a refined search around $b^{(2)}$ to obtain $b^{(1)}$.

5. At Level-0, perform a final refined search around $b^{(1)}$ to obtain the optimal biases $b_{\text{acc}}^{(0)}$ and $b_{\text{gyr}}^{(0)}$.

6. Compute the final bias $b^*$ and matching length $l^*$.

7. Align the source and target sequences using $b^*$ and $l^*$.

**Implementation Details**

In our implementation, we set the downsampling factors to $s_1 = 10$ and $s_2 = 10$, resulting in Level-1 and Level-2 sequences downsampled by factors of 10 and 100, respectively.

The search ranges at each level are defined as:

$$\text{Level-2: } b \in [-b_{\text{max}}, b_{\text{max}}], \quad b_{\text{max}} = 100, \tag{46}$$

$$\text{Level-1: } b \in [b^{(2)} - 10s_1, b^{(2)} + 10s_1], \tag{47}$$

$$\text{Level-0: } b \in [b^{(1)} - 10s_0, b^{(1)} + 10s_0], \tag{48}$$

where $s_0 = 1$ is the downsampling factor at Level-0 (original data).

**Computational Efficiency**

By employing the multi-level hierarchical search, we significantly reduce the computational complexity compared to an exhaustive search at the original sampling rate. At Level-2, the coarse search over a wide range of biases is feasible due to the reduced sequence length. The refined searches at higher resolutions are limited to small ranges around the biases found at lower levels, ensuring that the total computational cost remains manageable.

**Visualization of the Alignment Results**

Fig. 11, Fig. 12 and Fig. 11 showcase the IMU registration results for two trajectories. The high degree of overlap between the two IMU streams after alignment demonstrates the effectiveness of our proposed method.

# F   MORE VISUALIZATION RESULTS

**More Examples on Our SEE-600K Dataset**

The additional visualizations provided in Fig. 14 and Fig. 15 demonstrate the diversity of the SEE-600K dataset. The dataset captures a wide variety of scenes, both indoors and outdoors, including objects like plants, buildings, and everyday items. This diversity reflects common real-world scenarios, ensuring comprehensive coverage of typical environments. The images span different lighting conditions, showcasing the dataset's ability to handle various illumination levels, from low to high light.

**More Visualization on SEE-600K Dataset**

Fig.16,17,18,19,20,21 showcase additional visual results on the SEE-600K dataset. These examples further demonstrate the robustness and consistency of our proposed SEE-Net method. Notably, when using a brightness prompt of $0.5$, SEE-Net is capable of generating more stable and higher-quality images. In some cases, the output even surpasses the quality of the ground truth normal-light image (GT), showing the strength of our approach in various lighting conditions.

Additionally, it's important to highlight certain challenging cases, as shown in Fig. 20. For instance, in regions with highly detailed textures or areas requiring high-resolution recovery, all current methods, including ours, struggle to achieve optimal results. Despite this, SEE-Net continues to show relatively better performance compared to existing methods, particularly in maintaining image quality and stability. These results illustrate the potential of our method to handle complex scenarios, but they also indicate areas where further improvements could be made in future research.

By highlighting both the strengths and limitations of our approach, these visualizations provide valuable insights into the practical capabilities of SEE-Net across a wide range of real-world lighting conditions and complex scenes.

**More Visualization on SDE Dataset**

Fig. 23 and 24 present additional visualizations from the SDE dataset, specifically focusing on challenging low-light outdoor scenes. These low-light environments often come with significant noise, which poses a substantial challenge for current low-light enhancement methods. Our method demonstrates stable performance in addressing these noisy scenes, effectively enhancing the image quality while mitigating the noise, thereby highlighting the robustness of our approach in handling complex low-light conditions.

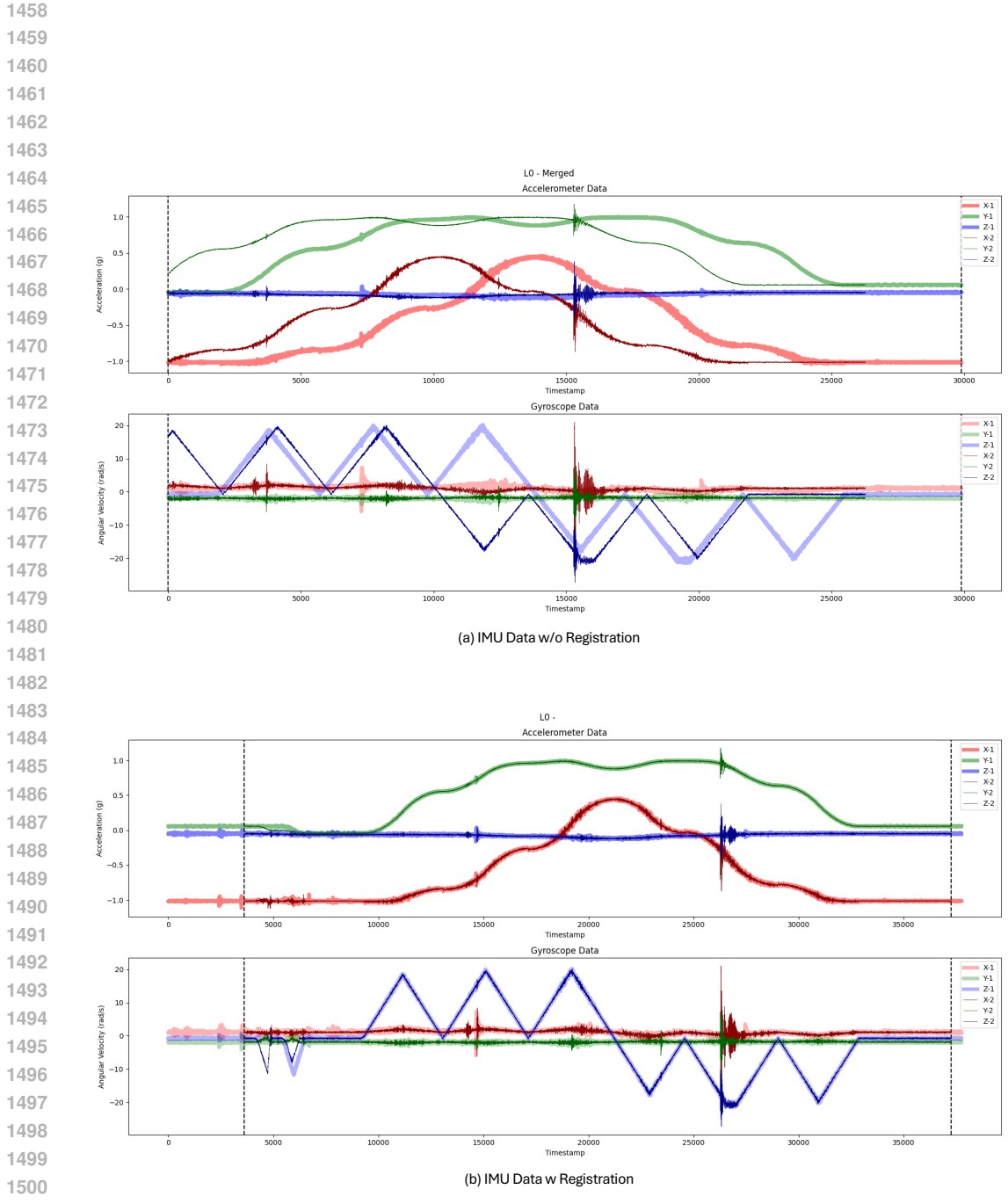

Figure 11: Original IMU data and registered IMU data.

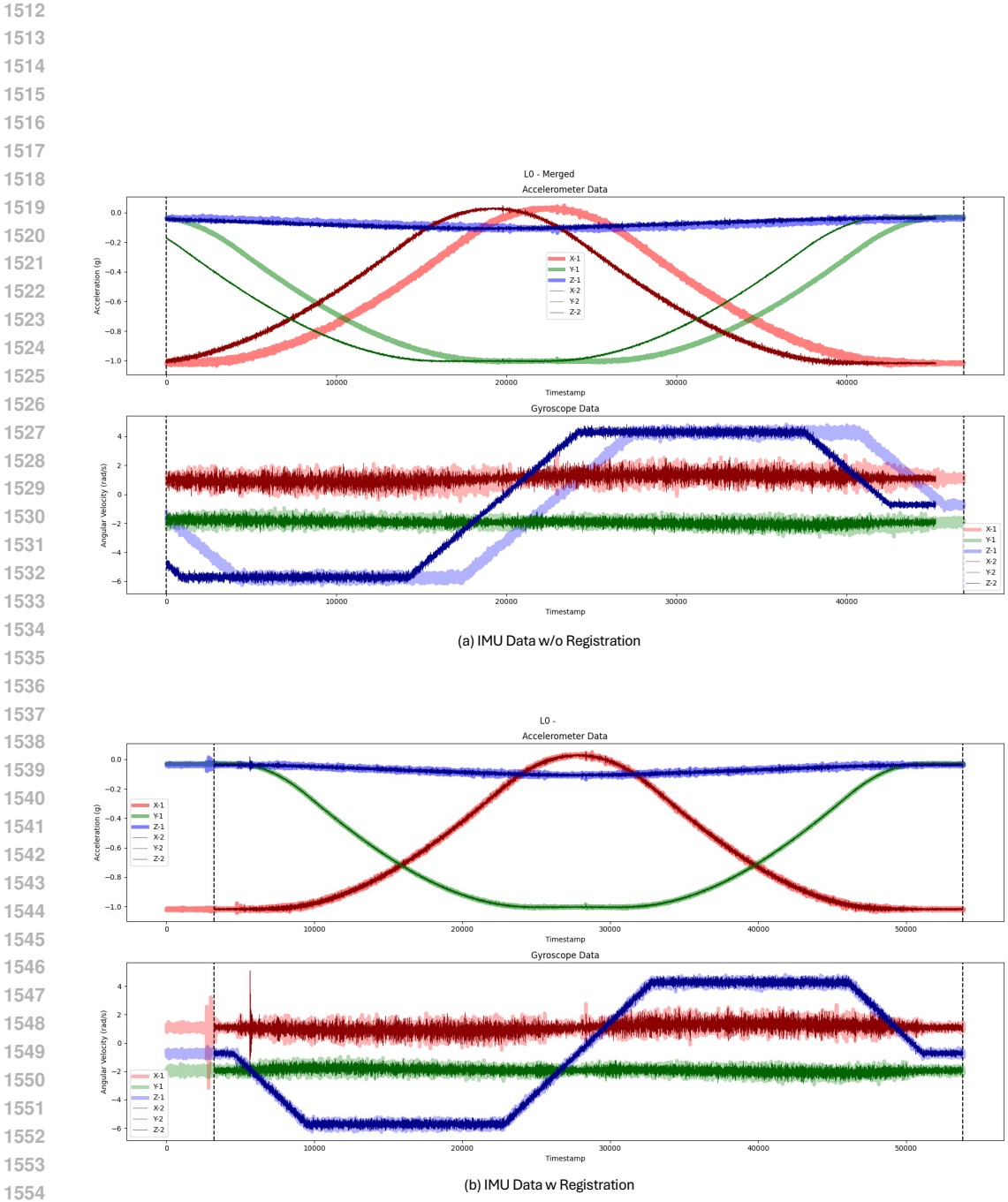

Figure 12: Original IMU data and registered IMU data.

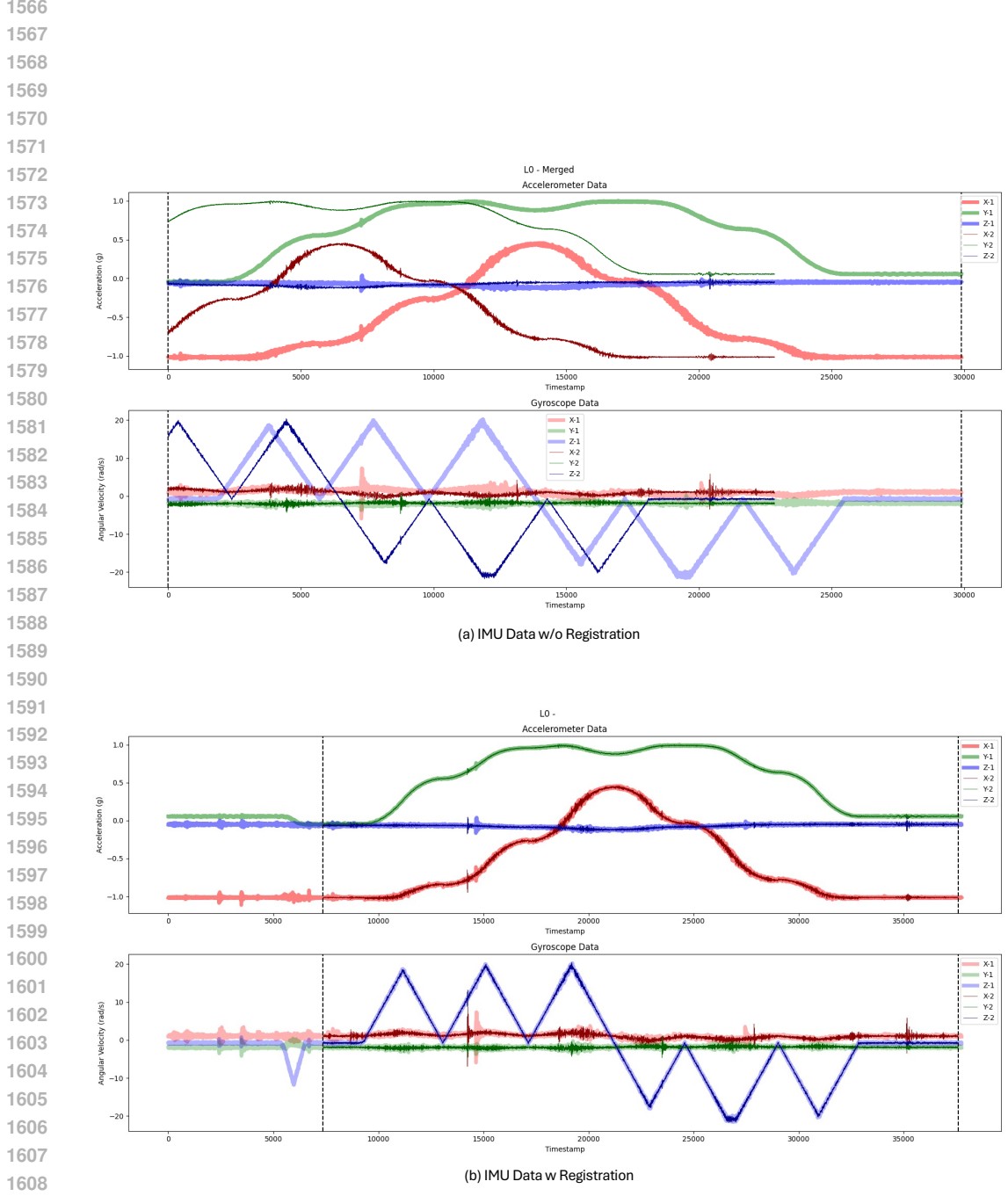

Figure 13: Original IMU data and registered IMU data.

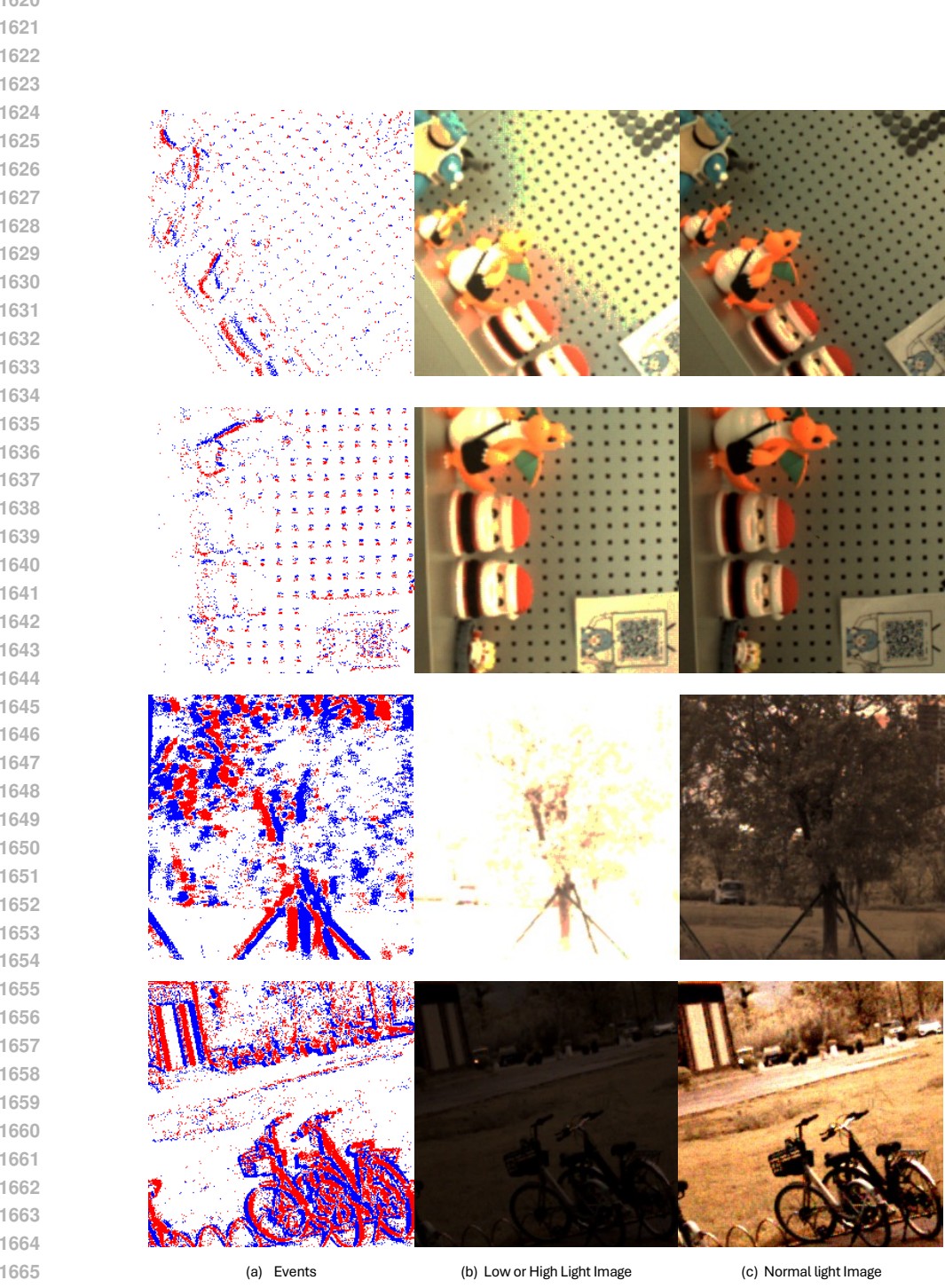

     (a)  Events         (b) Low or High Light Image        (c) Normal light Image

Figure 14: More examples on our SEE-600K dataset.

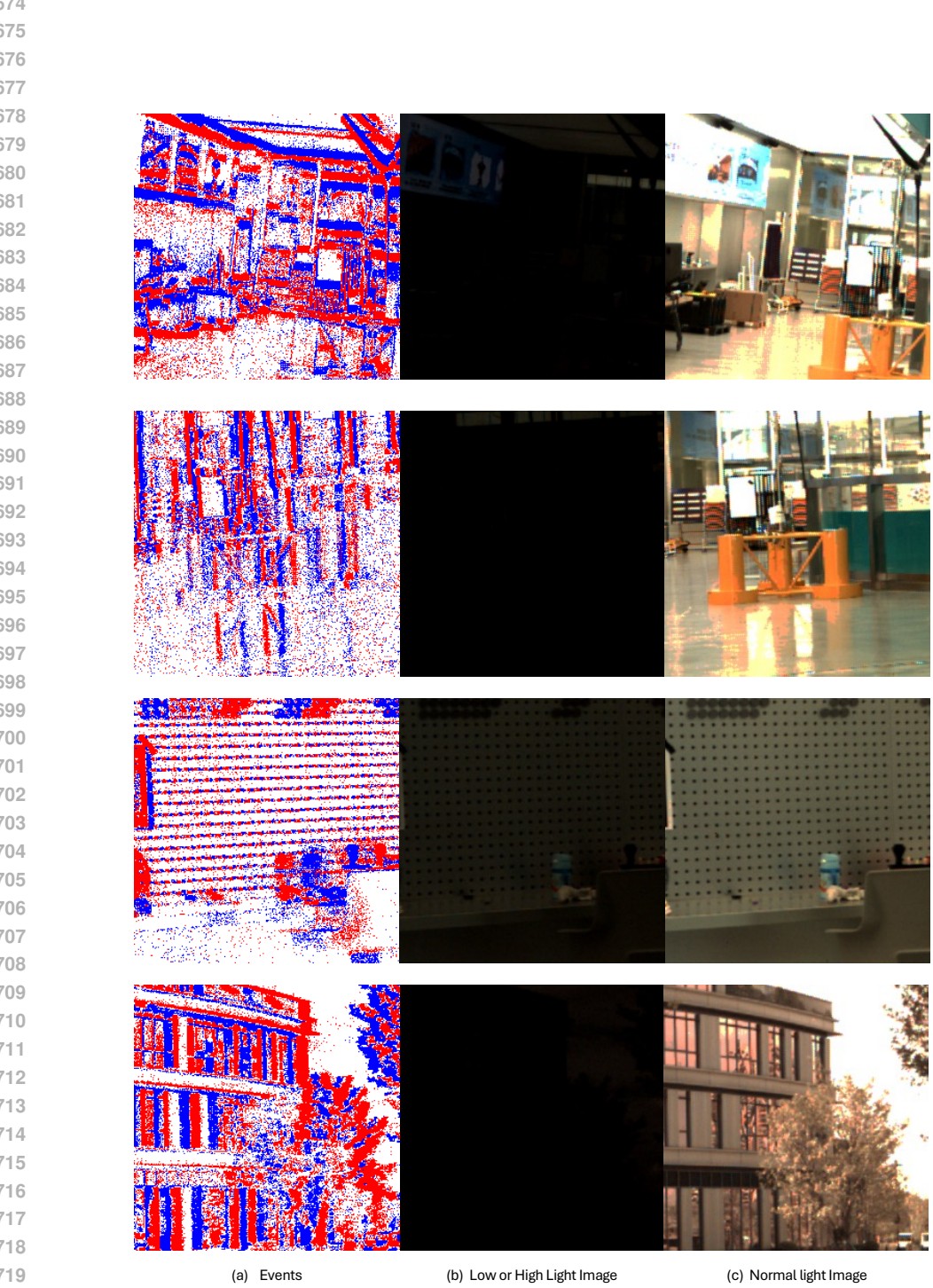

(a) Events        (b) Low or High Light Image        (c) Normal light Image

Figure 15: More examples on our SEE-600K dataset.

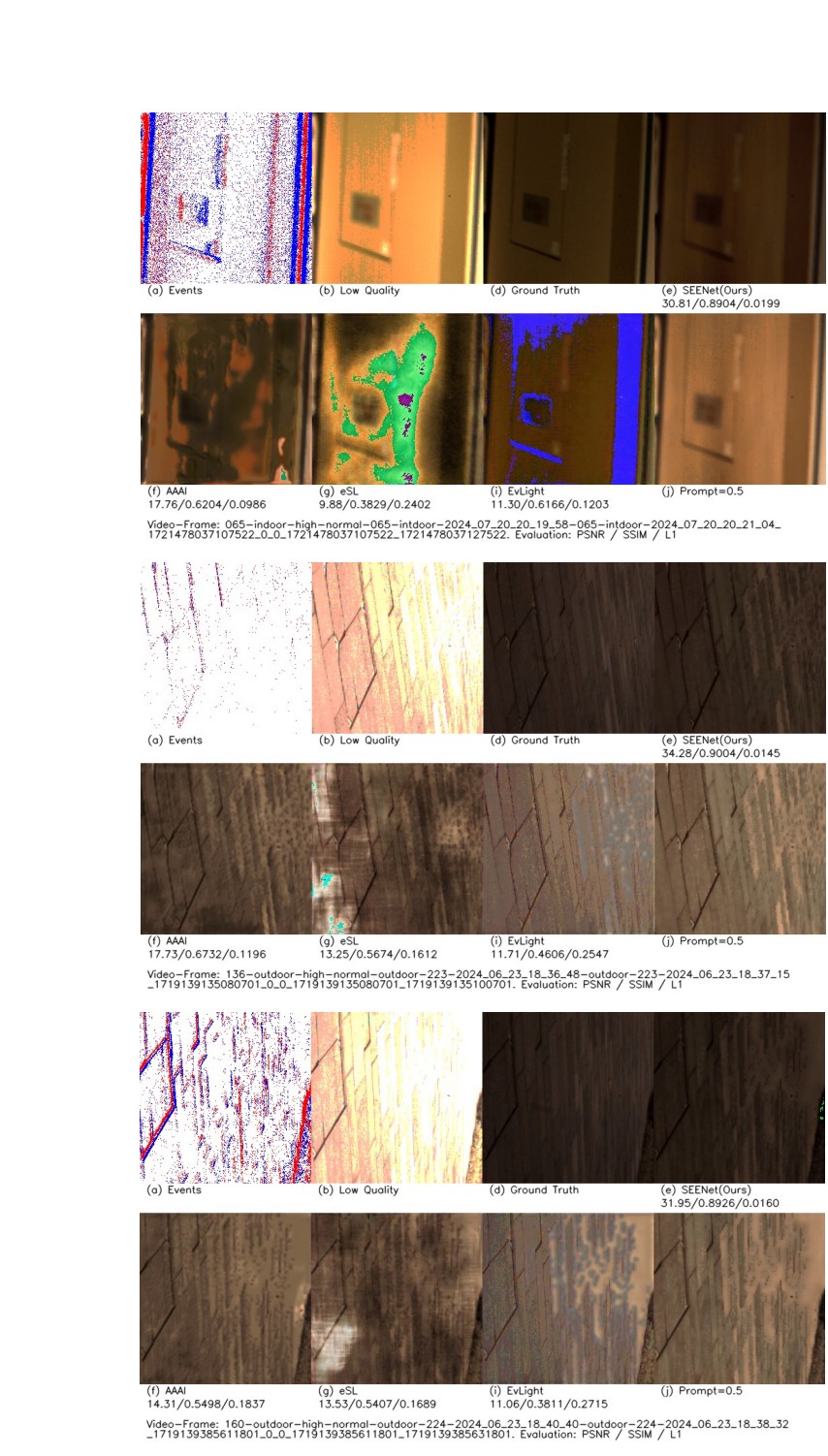

Figure 16: More visualization results on SEE-600k dataset.

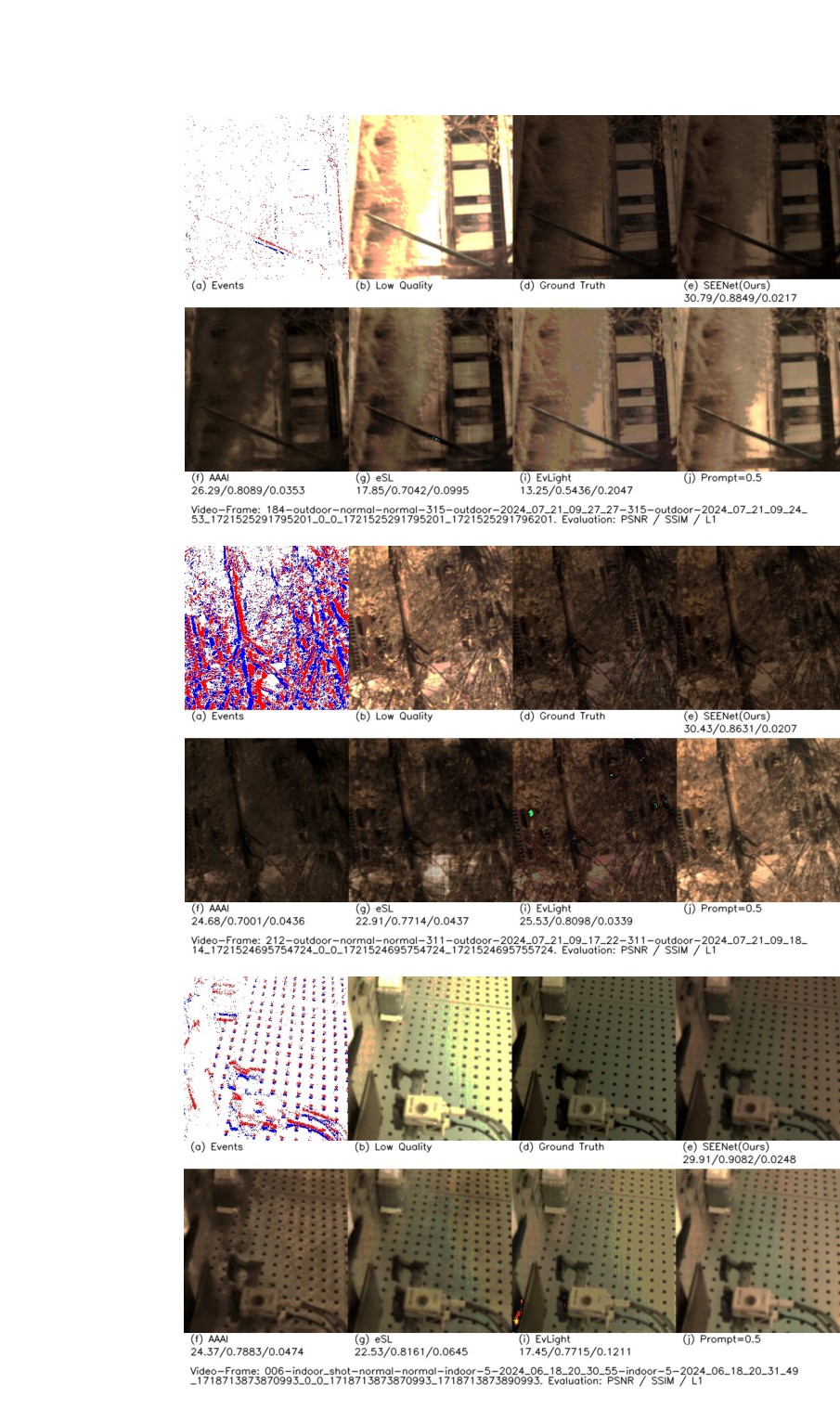

Figure 17: More visualization results on SEE-600k dataset.

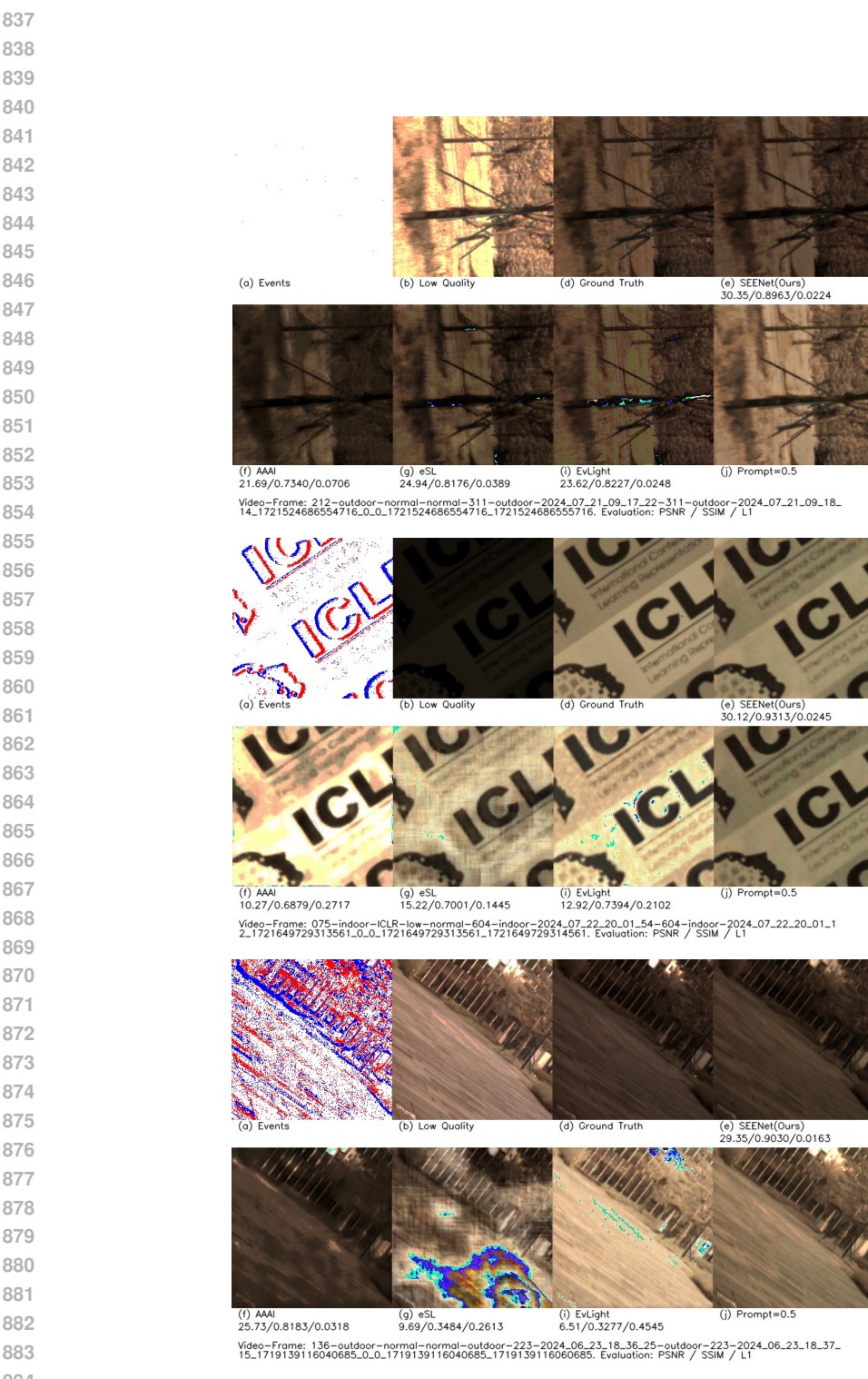

Figure 18: More visualization results on SEE-600k dataset.

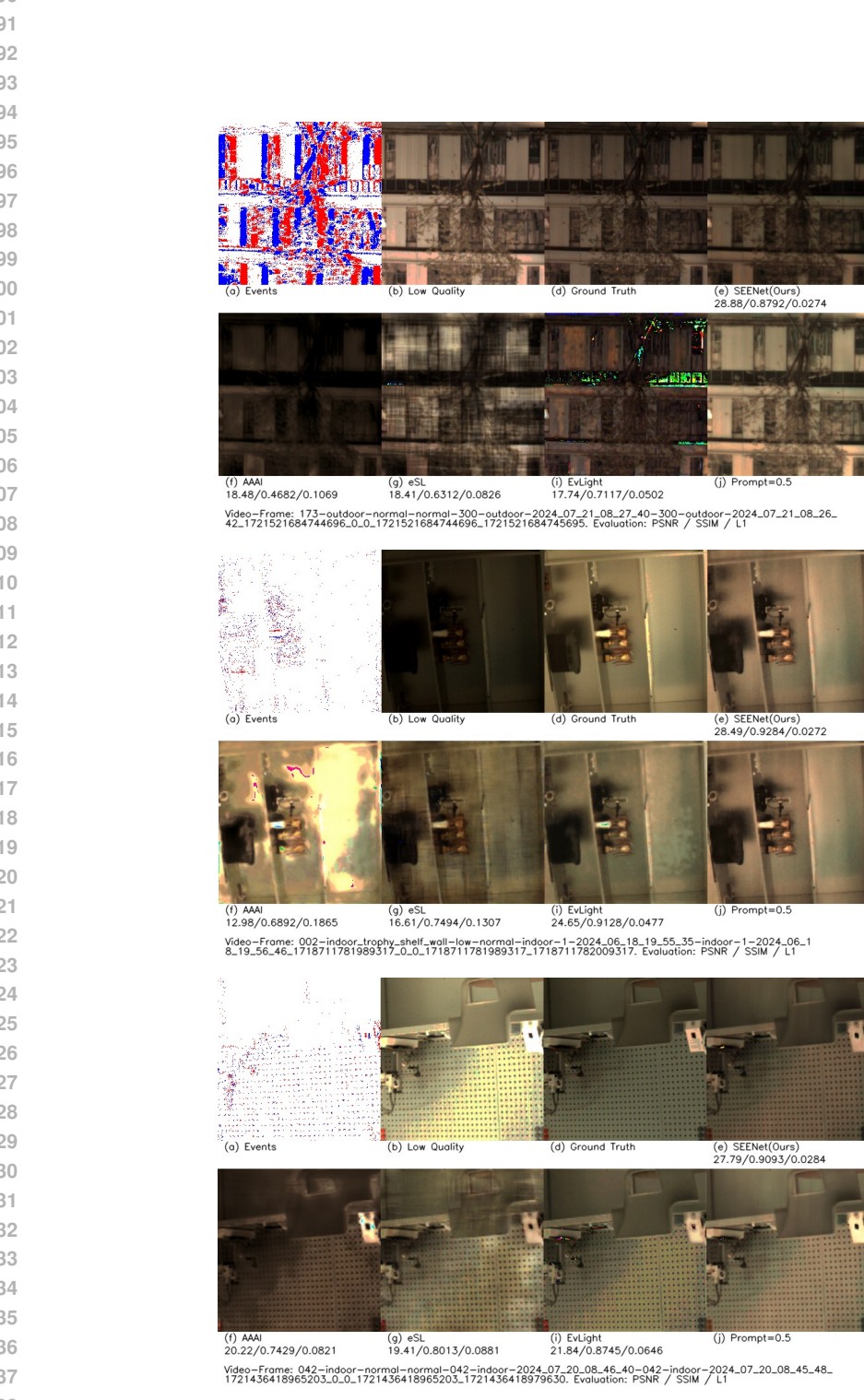

Figure 19: More visualization results on SEE-600k dataset.

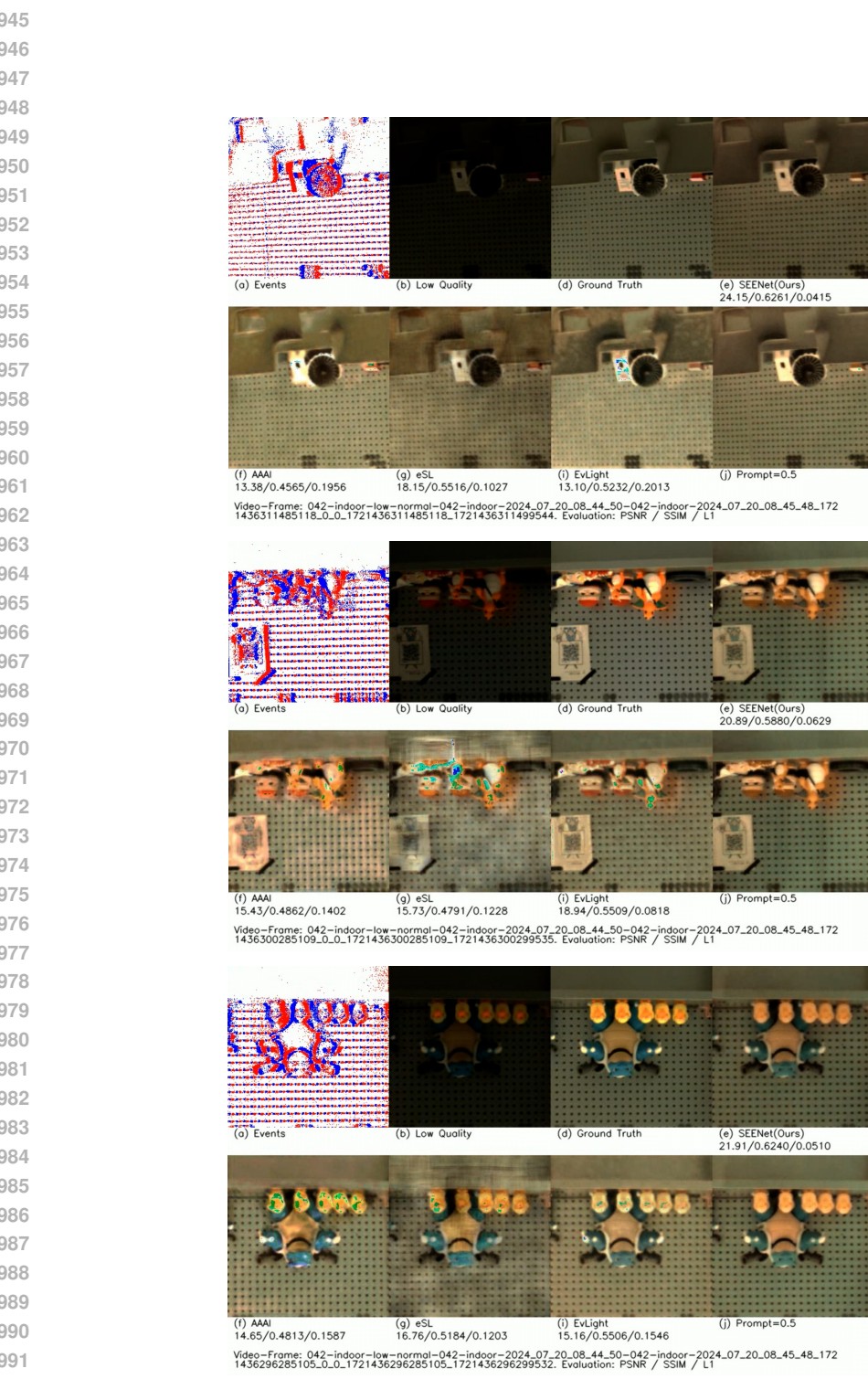

Figure 20: More visualization results on SEE-600k dataset.

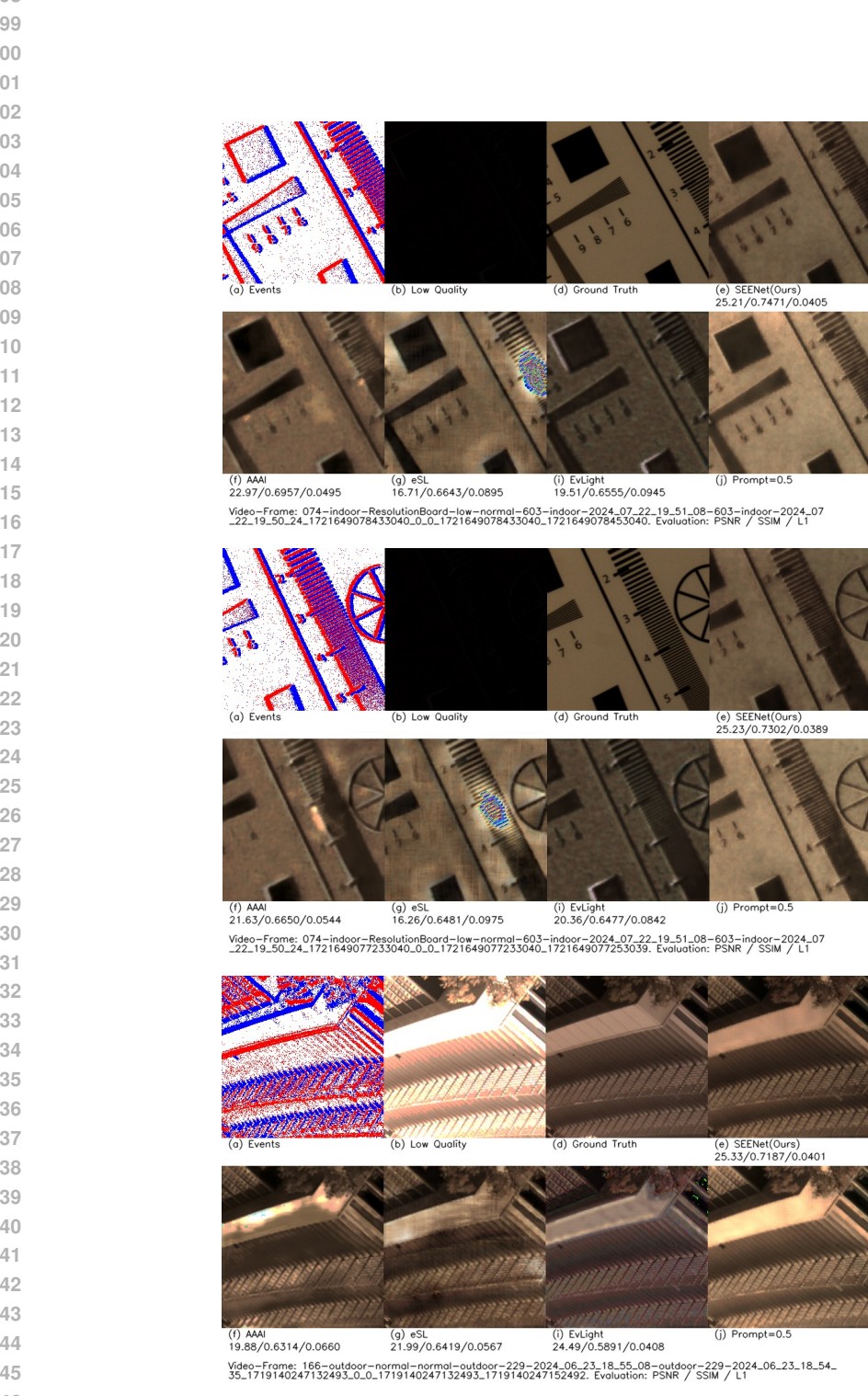

Figure 21: More visualization results on SEE-600k dataset.

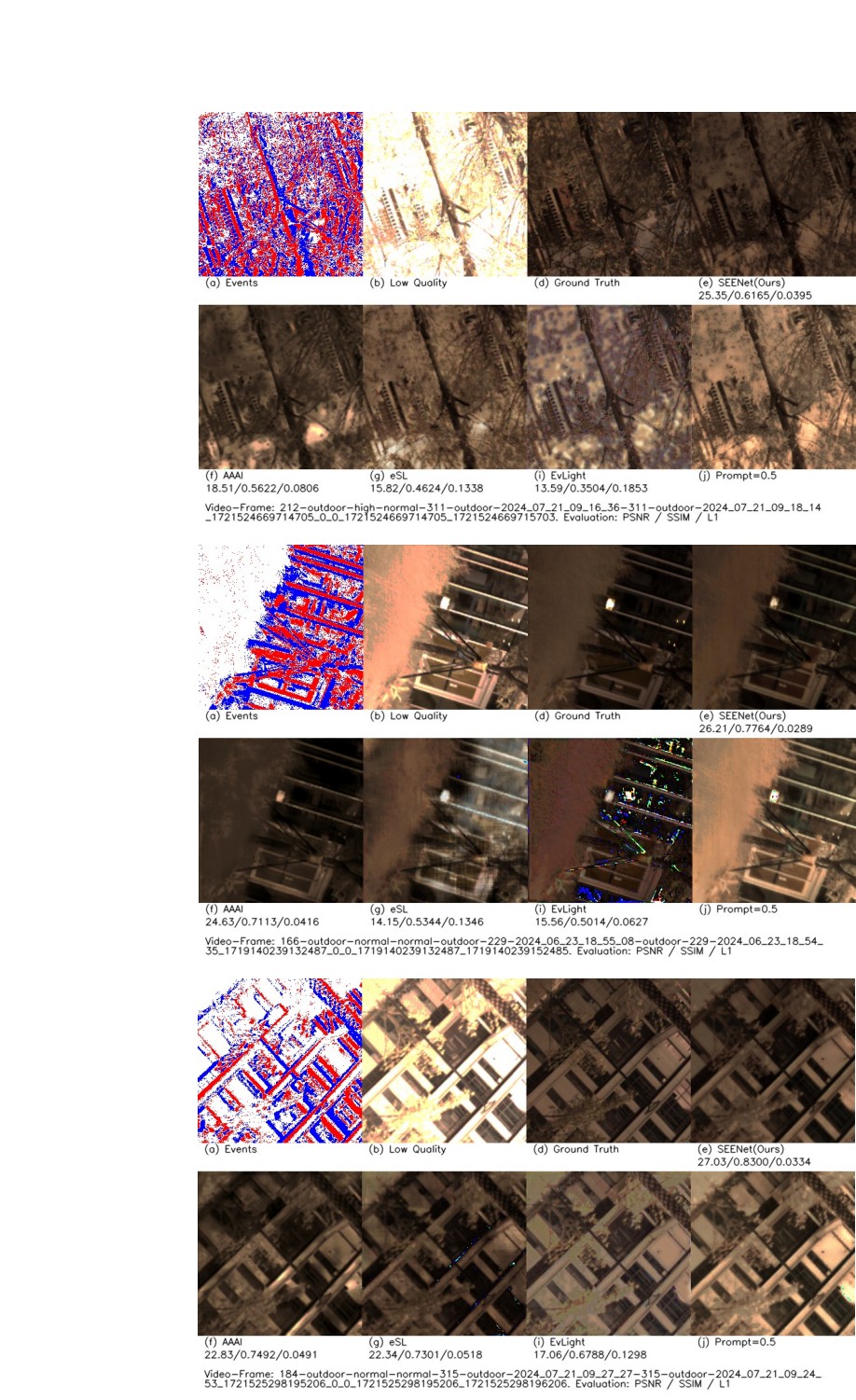

Figure 22: More visualization results on SEE-600k dataset.

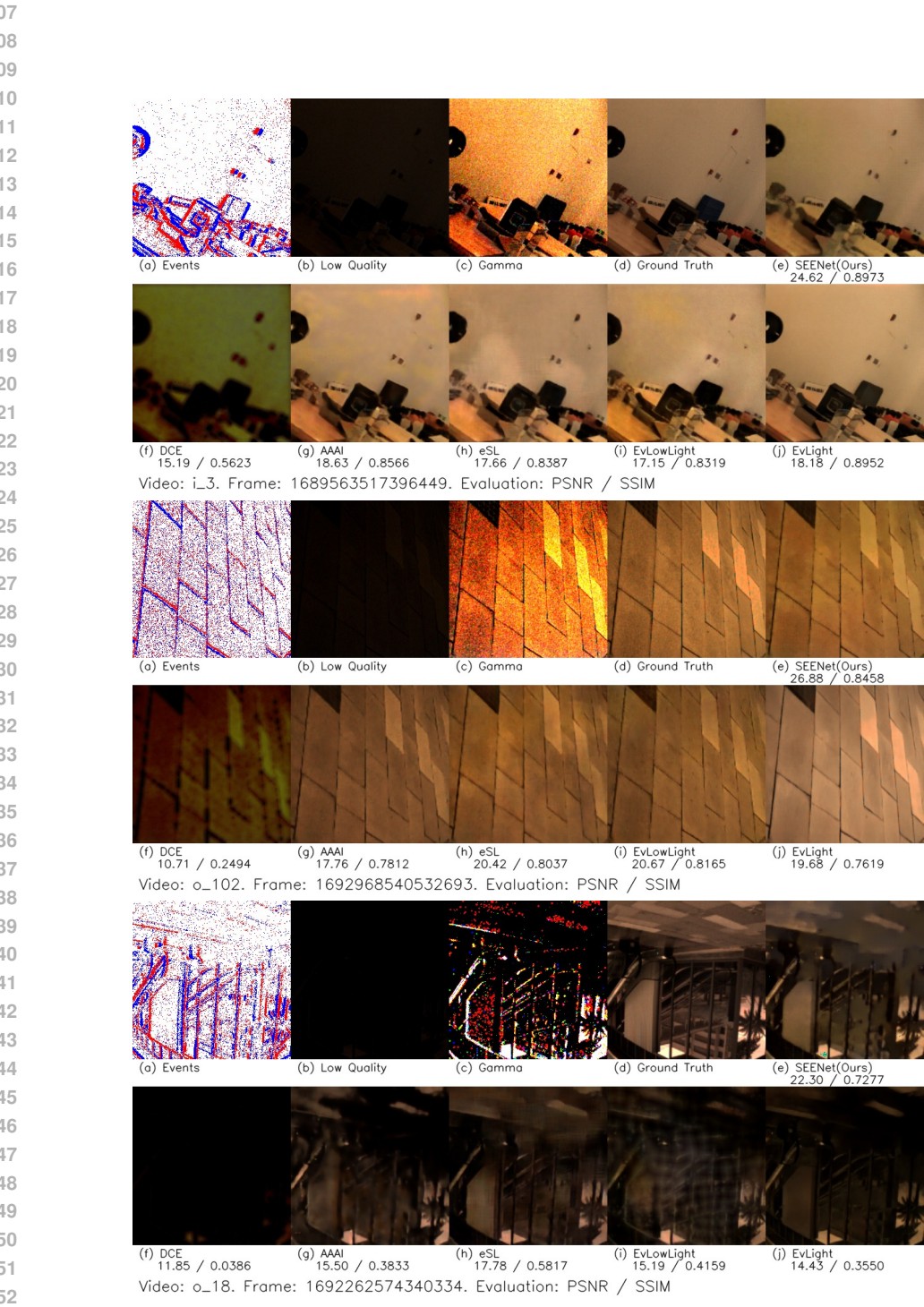

Figure 23: More visualization results on SDE dataset.

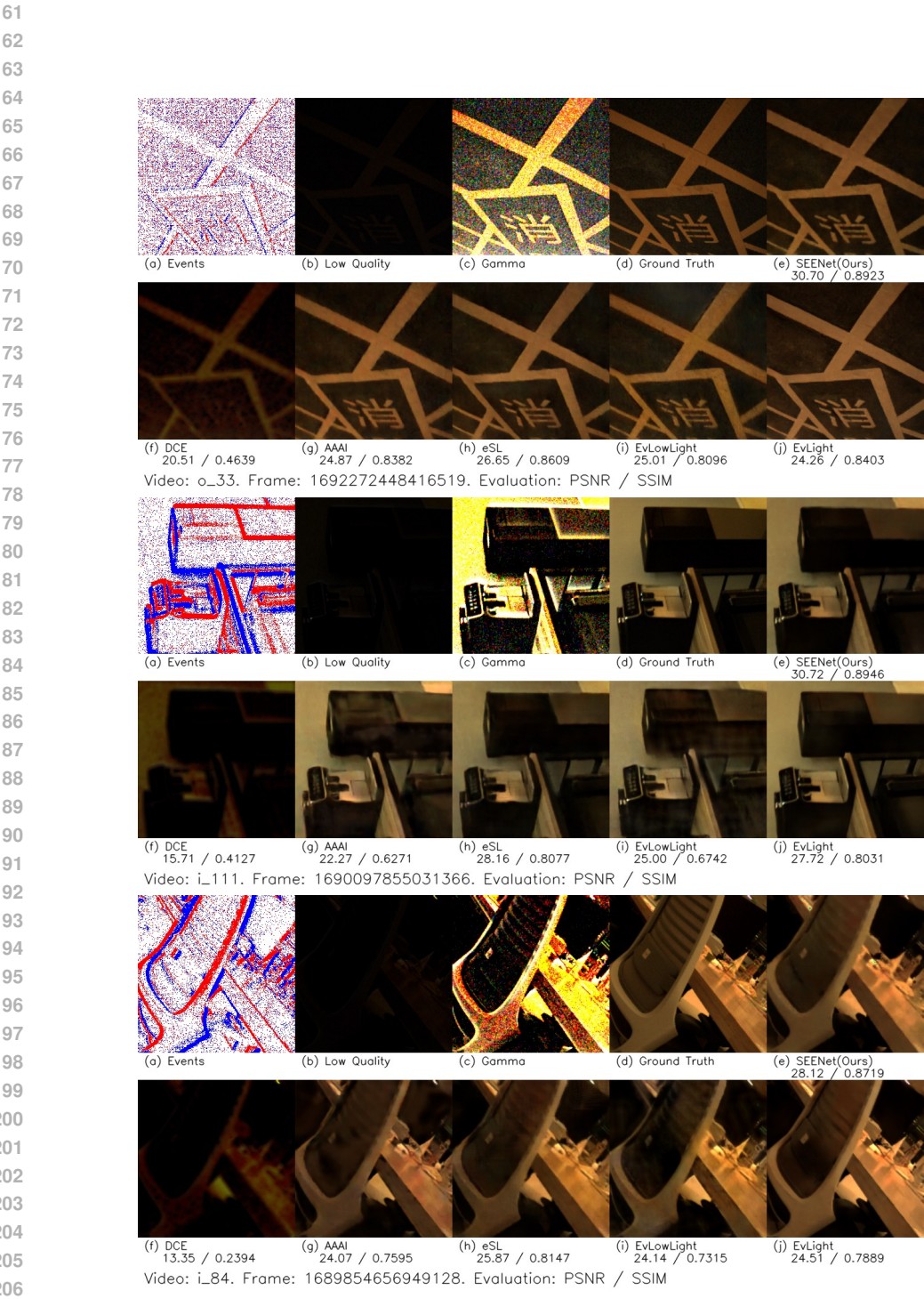

Figure 24: More visualization results on SDE dataset.

