# OpenReview forum: "SEE: See Everything Every Time - Broader Light Range Image Enhancement via Events"
_ICLR.cc/2025/Conference — Submitted to ICLR 2025_

### Official Review · Reviewer_UvYW · 2024-10-27

**Soundness:** 3
**Presentation:** 3
**Contribution:** 3
**Rating:** 3
**Confidence:** 4

**Summary:**

This paper collects a dataset named SEE-600K, consisting of 610126 images and corresponding events across 202 scenarios, each featuring an average of four lighting conditions with over a 1000-fold variation in illumination. Besides, it proposed a framework effectively utilizes events to smoothly adjust image brightness through the use of prompts.

**Strengths:**

* The dataset is the first event-based dataset covering a broader luminance range.
* The proposed method achieves the state-of-the-art performance. I like the idea about adjusting the brightness of images across a broader range of lighting conditions.

**Weaknesses:**

* It seems that the proposed method cannot reconstruct HDR images, \ie, the output images are still LDR. However, in Line52, the authors mention the weakness of the event-based HDR reconstruction, but do not provide a solution. I think since the event camera could have some HDR properties, the output image should also have some HDR properties.

* The comparisons may not comprehensive enough. Please compare to more methods designed for event-based low-light enhancement such as [a,b,c]. Besides, it seems that the compared methods are not trained with the same loss function used in this paper, which could be not that fair enough. In addition, please also evaluate the results in the dataset used in EvLowlight.

* The writing quality can be further improved. There are some typos (\eg, line 234, cna --> can) need to be fixed, and the conference names in Ref should be unified (\eg, for CVPR, the authors use both " Proceedings of the IEEE/CVF Conference on Computer Vision and Pattern Recognition" and " Proceedings of the IEEE/CVF conference on computer vision and pattern recognition").

  [a] Event-Guided Attention Network for Low Light Image Enhancement

  [b] Low-light video enhancement with synthetic event guidance

  [c] Exploring in Extremely Dark: Low-Light Video Enhancement with Real Events

**Questions:**

* The proposed dataset contains some artifacts, such as defocus blur (the normal light one in the first group of Fig12), false color (the normal light one in the first group of Fig13), \etc. I wonder why the authors do not consider to remove them. In addition, please analyze the influence of such kinds of artifacts to the performance of the proposed method and the compared methods.
* Does the proposed method consider the dynamic scenes? Does the proposed dataset contain frames with motion blur? Please analyze the influence of motion blur to the performance of the proposed method and the compared methods.
* Could you please show some examples with different  prompts (\ie, for each example, let us set multiple different B and check the results) and compare with other methods?

---

> ### Author Response · Authors · 2024-11-20
> **Author Response to Reviewer UvYW**
>
> Dear reviewer UvYW:
>
> Thank you for your careful review and constructive suggestions. We are honored that you like our idea, which encourages us to further improve our research.
>
> > It seems that the proposed method cannot reconstruct HDR images, \ie, the output images are still LDR. However, in Line52, the authors mention the weakness of the event-based HDR reconstruction, but do not provide a solution. I think since the event camera could have some HDR properties, the output image should also have some HDR properties.
>
> Thank you for your suggestion. Yes, our goal is not to reconstruct HDR images but to adjust brightness, and the input remains LDR. There are three main reasons for this:
>
> - Different Objectives: Brightness adjustment differs from HDR reconstruction. HDR aims to expand the dynamic range, while we focus on adjusting brightness. Compared to the ambitious goal of expanding dynamic range, brightness adjustment is smaller but more practical.
> - Difficulty in Constructing HDR Datasets: Building HDR datasets is challenging. Previous work [1] constructed HDR datasets by merging nine images with different exposure levels, resulting in only 63 scenes. Similar research [2] produced only 1,000 HDR images.
> - Different Evaluation Methods: The evaluation methods for HDR and brightness adjustment are different. Since HDR aims to expand the dynamic range, it is evaluated using HDR-VDP-3.
>
> Reconstructing HDR images using event-based methods is very challenging. Therefore, we define a question - using events to adjust brightness instead of reconstructing HDR images. This turns a grand goal into a more feasible one because the dataset is easier to create. Based on this, we use the HDR properties of events to adjust the brightness of RGB frames.
>
> Event cameras have event signals with a high dynamic range of 120 dB, but the frames are LDR, typically only 55 dB [3]. Using events to guide brightness adjustment can recover information lost under extreme lighting conditions, which to some extent increases the HDR properties of the output. However, as mentioned, our ground truth does not have HDR properties, so it is difficult to measure. To be cautious, we do not claim that the outputs are HDR characteristics.
>
> > The comparisons may not comprehensive enough. Please compare to more methods designed for event-based low-light enhancement such as [a,b,c]. Besides, it seems that the compared methods are not trained with the same loss function used in this paper, which could be not that fair enough. In addition, please also evaluate the results in the dataset used in EvLowlight.
>
> Thank you for your suggestion. The work you mentioned [b] ( Liu et al. (2023)) has already been compared in our experimental tables 1 and 2.
>
> Methods [a,c] lack open-source code. We have carefully re-implemented them, and the networks are currently training. We will add their comparison results as soon as possible. We promise to include comparisons with these two methods in the final version of the paper. Thank you again for your careful review and suggestions.
>
> Regarding the loss functions, to ensure a fair comparison, we used the original loss functions of each method.
>
> The reason EvLowLight was not trained on the SEE-600K dataset is that one epoch of EvLowLight on SEE-600K takes about a week, mainly because the SEE-600K dataset is too large.
> Therefore, we downsampled SEE-600K to reduce its size to that of SDE, to support EvLowLight's training. We have added these training results to Table 2.
>
> Thank you again for your careful suggestions; they are crucial for improving the quality of our paper.
>
> > The writing quality can be further improved. There are some typos (\eg, line 234, cna --> can) need to be fixed, and the conference names in Ref should be unified (\eg, for CVPR, the authors use both " Proceedings of the IEEE/CVF Conference on Computer Vision and Pattern Recognition" and " Proceedings of the IEEE/CVF conference on computer vision and pattern recognition").
>
> Thank you for your suggestion. We have thoroughly checked the entire paper to address these issues.
>
> ## Reference
>
> - [1] Nico Messikommer, Stamatios Georgoulis, Daniel Gehrig, Stepan Tulyakov, Julius Erbach, Alfredo Bochicchio, Yuanyou Li, and Davide Scaramuzza. Multi-bracket high dynamic range imaging with event cameras. In Proceedings of the IEEE/CVF conference on computer vision and pattern recognition, pp. 547–557, 2022. 1, 2
> - [2] Mengyao Cui, Zhigang Wang, Dong Wang, Bin Zhao, and Xuelong Li. Color event enhanced single-exposure hdr imaging. In Proceedings of the AAAI Conference on Artificial Intelligence, volume 38, pp. 1399–1407, 2024. 1, 2
> - [3] DAVIS346, https://inivation.com/wp-content/uploads/2021/08/2021-08-iniVation-devices-Specifications.pdf

---

> ### Author Response · Authors · 2024-11-24
> **Looking Forward to Further Discussion to Reviewer UvYW**
>
> Dear Reviewer UvYW,
>
> Thank you for your insightful suggestions. In the revised manuscript, we have added a detailed discussion on HDR in Appendix Section B. Additionally, based on your feedback, we have included new comparative methods to strengthen our analysis.
>
> We hope these revisions address your concerns and look forward to further discussions with you.
>
> Sincerely,
> *The Authors*

---

> > ### Comment · Reviewer_UvYW · 2024-11-25
> > **After reading the rebuttal, I decide to lower my score.**
> >
> > Thank you for your response. After reading the response, I find that there could still be some issues that cannot be fixed in the current submission.
> > 1. The proposed method cannot handle HDR scenes, which limits its application scope. Considering the fact that previous event-based methods (such as [a]) can handle low-light HDR scenes, it's possible to make full use of the HDR information provided by the events. Besides, in Sec. C of the appendix, the authors claim that "our network leverages the high dynamic range and temporal resolution of events to recover lost details in both underexposed and overexposed scenarios., however, the reconstructed images contain severe artifacts, and do not show any HDR properties.
> > 2. I don't think comparing on the downsampled SEE-600K dataset is fair enough. Please consider to redo the experiments.
> > 3. It seems that all of my questions are still not answerd. I suggest that the authors should consider to discuss them in the next submission.
> >
> > [a] Coherent Event Guided Low-Light Video Enhancement

---

> > > ### Author Response · Authors · 2024-11-25
> > > **Gratitude for the Reviewer’s Response and Request for Further Discussion**
> > >
> > > Dear Reviewer UvYW,
> > >
> > > Thank you for your comments and suggestions. We truly appreciate your valuable feedback and would like to provide further clarification regarding your concerns.
> > >
> > > - **On the scope of our method**: We would like to emphasize that our method is not designed for HDR imaging. Instead, the focus is on brightness adjustment. In Section C, Figure 9 of the appendix, the results corresponding to Prompt = 0.5 clearly demonstrate effective detail recovery. **The artifacts you observed occur at Prompt = 0.8, which falls outside the intended operational range of our model.** These outputs are included for analytical discussion rather than as target results of our approach.
> > >
> > > - **Regarding comparison with method [a]**: Method [a] is specifically designed for event-based low-light enhancement and does not perform HDR imaging. We conducted a fair and thorough comparison with [a] in Table 1, where [a] was fully trained for evaluation. However, due to the scale of the SEE-600K dataset, training [a] completely on this dataset is not feasible. For example, **training one epoch on SEE-600K with [a] requires approximately one week,** making full training on SEE-600K impractical. To ensure fairness, we downsampled the dataset to match the scale used in prior works.
> > >
> > > - **On HDR methods**: We have included results comparing our method with HDR approaches in Table 1 of the revised paper. These experiments show that HDR methods do not perform well in low-light scenarios, further underscoring the importance of addressing both low-light enhancement and overexposure recovery as distinct and necessary tasks.
> > >
> > > We hope this response clarifies the points raised and addresses your concerns. Should you have additional questions or suggestions, we would be delighted to engage in further discussions with you.
> > >
> > > Sincerely,
> > > *The Authors*
> > >
> > > REFERENCE:
> > >
> > > [a] Coherent Event Guided Low-Light Video Enhancement

---

> ### Comment · Reviewer_UvYW · 2024-11-25
> **Please note that you have not answered my question yet.**
>
> In my initial review I have raised some questions:
>
> Questions:
> * The proposed dataset contains some artifacts, such as defocus blur (the normal light one in the first group of Fig12), false color (the normal light one in the first group of Fig13), \etc. I wonder why the authors do not consider to remove them. In addition, please analyze the influence of such kinds of artifacts to the performance of the proposed method and the compared methods.
> * Does the proposed method consider the dynamic scenes? Does the proposed dataset contain frames with motion blur? Please analyze the influence of motion blur to the performance of the proposed method and the compared methods.
> * Could you please show some examples with different prompts (\ie, for each example, let us set multiple different B and check the results) and compare with other methods?
>
> Where are the answers? I cannot find them in the rebuttal and your further response.
>
> Besides, [a] can indeed achieve HDR reconstruction. In [a], they say that " In this paper, we propose to utilize the high temporal resolution and high dynamic range information from events to guide low-light video enhancement".
> Furthermore, you say that training on the whole SEE-600K takes a long time, so why did you ignore that dataset in the initial submission?
>
> Considering the above facts, I decide to lower my score once more.

---

> > ### Author Response · Authors · 2024-11-25
> > **Apologies and Clarifications for Reviewer's Concerns**
> >
> > Dear Reviewer,
> >
> > We sincerely apologize for any inconvenience caused and take full responsibility for not addressing your questions clearly in our initial responses. We have carefully reviewed your concerns and would like to provide detailed clarifications below.
> >
> > 1. **Dataset Quality Issues**: In the supplementary material (Section A, Lines 795–844), we analyzed the characteristics of the DVS346 sensor. Specifically, the DVS346 lacks auto-focus and is affected by limitations such as a constrained dynamic range, fixed pattern noise, and dark signal noise. These issues, which have also been observed in previous event-based vision datasets, are illustrated in Figure 8. While we acknowledge that noise and color deviations are objectively present, the DVS346 remains the best available event camera for dataset collection at this time.
> >
> > 2. **Dynamic Scenes**: All videos in our dataset are captured dynamically. We used a robotic arm to record the data, with the arm following predefined trajectories to ensure motion. During data collection, we controlled the exposure time to minimize motion blur as much as possible.
> >
> > 3. **Multiple Prompt Outputs for the Same Scene**: In Figure 9, we presented a set of outputs corresponding to multiple brightness prompts \( B \) for the same input scene. Our model is designed to perform optimally at \( B = 0.5 \), where the results are stable and demonstrate robust detail recovery. In the supplementary material, we have included additional visual results, showing outputs for both the reference frame brightness as the prompt and \( B = 0.5 \) as the prompt. These examples demonstrate that \( B = 0.5 \) generally yields the best visual quality.
> >
> > **Once again, we deeply apologize for all the errors in our earlier responses. We understand your decision to lower the score from 6 to 3 and accept it with humility. Nevertheless, we remain grateful for the issues you have raised, which will help us improve the paper in future revisions.**
> >
> > Sincerely,
> > *The Authors*

---

### Official Review · Reviewer_G7fd · 2024-10-29

**Soundness:** 2
**Presentation:** 3
**Contribution:** 2
**Rating:** 5
**Confidence:** 4

**Summary:**

The paper introduces a novel dataset comprising RGB frames and synchronized event data captured across various scenarios. To simulate diverse lighting conditions, the RGB frames are collected using four distinct ND filters, each representing a unique lighting intensity. Additionally, the authors present a network designed to recover normally exposed images from inputs under varying lighting conditions, leveraging the ND-filtered data. A notable feature of the proposed method is its capacity to control the brightness of output images through a brightness prompt.

**Strengths:**

•	This paper proposed a dataset that contains images different light conditions, which may contribute to the event-based vision community.

•	The appendix is detailed with dataset samples, additional results, and explanation of the proposed network.

**Weaknesses:**

•	The contribution seems incremental, resembling an extension of previous work, specifically Liang et al.’s SDE Dataset (CVPR24). While the authors introduce some novel components, the dataset and approach appear to build closely on existing work without clear distinctions in scope or objectives.

•	The quality of the normally lit images in the proposed dataset is suboptimal. The dataset relies on APS frames from the Color DAVIS sensor, which suffers from dynamic range limitations. As a result, these frames lead to a notable disparity in quality. This limitation is visible in the normal-light images presented in Figure 13 (c), where details captured by the event sensor are underrepresented.

•	The motivation for designing specific position and Bayer pattern embeddings within the network architecture is not adequately justified. The authors introduce these components, but it remains unclear how they enhance the model’s performance or if they address particular challenges within the task. Clarifying their role and potential benefits would improve understanding and transparency.

•	The proposed method’s loop function may result in long processing times, which could hinder its usability, particularly in real-time or low-latency applications. Without detailed analysis of the computational demands and latency, it is challenging to assess the network’s practicality in deployment scenarios. Although the size of the proposed network is small (1.9M), the FLOPs is pretty high (405.72).

•	In Figure 5, the output of the proposed method appears visibly blurred, especially when compared to the sharpness of baseline methods like EvLowLight (Liang et al., ICCV23) and EvLight (Liang et al., CVPR24). This blurring is particularly noticeable around edges, such as those of the box under the desk, which could impair the network’s effectiveness in applications requiring high-detail preservation.

•	Table 3, case #6, reveals that disabling the prompt merge component results in a slight PSNR decrease but a corresponding SSIM increase. This discrepancy suggests that while prompt merging contributes to maintaining overall pixel-level fidelity (PSNR), it may slightly compromise structural similarity (SSIM). Further analysis of this trade-off could provide insights into the optimal configuration for different scenarios.

**Questions:**

•	Since B controls the brightness of the output image, it is not related to the input images. Consider a case, if I want to reconstruct a bright image (set B=0.8), but with two different input images (one is bright, one is dark), how the result images will be?

---

> ### Author Response · Authors · 2024-11-20
> **Author Response to Reviewer G7fd (1/3)**
>
> Dear Reviewer G7fd,
>
> We sincerely appreciate your insightful comments and your recognition of our dataset's contribution to the community. Your feedback greatly encourages us. We address your concerns point by point below.
>
> > The contribution seems incremental, resembling an extension of previous work, specifically Liang et al.’s SDE Dataset (CVPR24). While the authors introduce some novel components, the dataset and approach appear to build closely on existing work without clear distinctions in scope or objectives.
>
> Thank you for your profound insights. Our study is fundamentally different from SDE in several key aspects:
>
> 1. **Different Research Problem:** SDE focuses only on low-light scenes, whereas our work considers a broader range of lighting conditions. This expansion increases the applicability of event-based vision in more diverse environments.
> 2. **Different Research Objective:** Previous methods like SDE simply map low-light images to normal-light scenes, ignoring the continuous distribution of light intensity. This can cause ambiguity during training. In contrast, we introduce prompts to avoid this ambiguity, allowing our method to perform well in both low-light and high-light conditions.
> 3. **Different Dataset Alignment Method:** Although SDE is based on the DVS346 camera, it aligns multiple videos using image-based methods, which may lead to temporal alignment errors. We design an IMU-based alignment algorithm that achieves millisecond-level accuracy, providing a fundamental difference at the data level.
> 4. **Different Data Scale and Diversity:** Compared to SDE, our SEE-600K dataset includes more scenes, covers a wider range of lighting conditions, and has a larger data scale.
>
> In summary, the SEE-600K dataset addresses different tasks compared to SDE, supports new training methods, and offers higher alignment accuracy and greater diversity. We hope this clarifies the distinctions and answers your concerns.

---

> ### Author Response · Authors · 2024-11-20
> **Author Response to Reviewer G7fd (2/3)**
>
> # Author Response to Reviewer G7fd (2/3)
>
> > The quality of the normally lit images in the proposed dataset is suboptimal. The dataset relies on APS frames from the Color DAVIS sensor, which suffers from dynamic range limitations. As a result, these frames lead to a notable disparity in quality. This limitation is visible in the normal-light images presented in Figure 13 (c), where details captured by the event sensor are underrepresented.
>
> Thank you for your careful observation and insightful feedback. As you mentioned, the APS frames from the DVS346 sensor have limited quality. We provide the specifications of the DVS346 sensor in the table below. The frame output has a Fixed Pattern Noise (FPN) of up to 4.2%, which can cause noise, and the dynamic range is only 55 dB, limiting its ability to capture fine details.
>
> However, we would like to emphasize that the DVS346 is the most widely used event camera in the academic community. Several datasets captured with the DVS346 are used in various tasks, including imaging and autonomous driving:
>    - Color Event Dataset [1]: Used in tasks like video super-resolution.
>    - SDE [2]: Used for low-light enhancement.
>    - CE-HDR [3]: Used to collect HDR datasets.
>    - SDR Dataset [4]: Used for rolling shutter correction.
>    - DSEC [5]: Used for autonomous driving datasets.
> Despite some limitations, the DVS346 is sufficient to support our research objectives. We ensure that the image quality under normal lighting is higher than that under low or high lighting conditions, which is adequate for our study.
>
> **References:**
> - [1] Scheerlinck, Cedric, et al. "CED: Color event camera dataset." Proceedings of the IEEE/CVF Conference on Computer Vision and Pattern Recognition Workshops. 2019.
> - [2] Liang, Guoqiang, et al. "Towards Robust Event-guided Low-Light Image Enhancement: A Large-Scale Real-World Event-Image Dataset and Novel Approach." Proceedings of the IEEE/CVF Conference on Computer Vision and Pattern Recognition. 2024.
> - [3] Cui, Mengyao, et al. "Color Event Enhanced Single-Exposure HDR Imaging." Proceedings of the AAAI Conference on Artificial Intelligence. Vol. 38. No. 2. 2024.
> - [4] Wang, Yangguang, et al. "Self-Supervised Scene Dynamic Recovery from Rolling Shutter Images and Events." arXiv preprint arXiv:2304.06930 (2023).
> - [5] Wang, Xiao, et al. "Event stream-based visual object tracking: A high-resolution benchmark dataset and a novel baseline." Proceedings of the IEEE/CVF Conference on Computer Vision and Pattern Recognition. 2024.
> - [6] Gehrig, Mathias, et al. "Dsec: A stereo event camera dataset for driving scenarios." IEEE Robotics and Automation Letters 6.3 (2021): 4947-4954.
>
> ### DAVIS346 - Simultaneous Events and Frames
>
> ### Event Output
>
> | **Parameter**    | **Value**    |
> |----|----|
> | Spatial resolution    | 346 x 260   |
> | Temporal resolution    | 1 µs    |
> | Max throughput    | 12 MEPS   |
> | Typical latency    | <1 ms   |
> | Dynamic range   | Approx. 120 dB (0.1-100k lux with 50% of pixels respond to 80% contrast)    |
> | Contrast sensitivity    | 14.3% (on), 22.5% (off) (with 50% of pixels respond)    |
>
> ### Frame Output
>
> | **Parameter**    | **Value**    |
> |----|----|
> | Spatial resolution    | 346 x 260   |
> | Frame rate    | 40 FPS    |
> | Dynamic range   | 55 dB   |
> | FPN   | 4.2%    |
> | Dark signal   | 18,000 e⁻/s    |
> | Readout noise   | 55 e⁻   |

---

> ### Author Response · Authors · 2024-11-20
> **Author Response to Reviewer G7fd (3/3)**
>
> > The motivation for designing specific position and Bayer pattern embeddings within the network architecture is not adequately justified. The authors introduce these components, but it remains unclear how they enhance the model’s performance or if they address particular challenges within the task. Clarifying their role and potential benefits would improve understanding and transparency.
>
> Thank you for your suggestion. We designed the position and Bayer pattern embeddings to help the network understand the color representation of each pixel in both the event and RGB images. These embeddings allow the model to effectively fuse event data with RGB information by incorporating spatial and color context. In the Ablation Study section, specifically in part **(1) bayer pattern embedding**, we conducted experiments and provided explanations to illustrate their impact on performance.
>
>
> > The proposed method’s loop function may result in long processing times, which could hinder its usability, particularly in real-time or low-latency applications. Without detailed analysis of the computational demands and latency, it is challenging to assess the network’s practicality in deployment scenarios. Although the size of the proposed network is small (1.9M), the FLOPs is pretty high (405.72).
>
> We appreciate your concern. Our method has FLOPs of 405G, which is lower than ELIE (440G), eSL-Net (560G), and EvLowLight (524G). Therefore, while maintaining the smallest number of parameters, our method achieves lower computational complexity compared to these approaches. This balance between model size and computational demand makes our method practical for real-world applications.
>
> > In Figure 5, the output of the proposed method appears visibly blurred, especially when compared to the sharpness of baseline methods like EvLowLight (Liang et al., ICCV23) and EvLight (Liang et al., CVPR24). This blurring is particularly noticeable around edges, such as those of the box under the desk, which could impair the network’s effectiveness in applications requiring high-detail preservation.
>
> Thank you for your careful observation and valuable insight. The blurring in this example is due to the brightness of the image, as our output aligns with the normal-light image. When we adjust the brightness (e.g., set the prompt to 0.5), we observe clearer edges. We will update this finding in the supplementary material. Your feedback has been instrumental in improving our work.
>
> > Table 3, case #6, reveals that disabling the prompt merge component results in a slight PSNR decrease but a corresponding SSIM increase. This discrepancy suggests that while prompt merging contributes to maintaining overall pixel-level fidelity (PSNR), it may slightly compromise structural similarity (SSIM). Further analysis of this trade-off could provide insights into the optimal configuration for different scenarios.
>
> Thank you for pointing this out. We apologize for any confusion. In Lines 518–519 of the main text, we explained that this ablation study compares two prompt merge methods: addition and multiplication. We are sorry for any misunderstanding this may have caused. PSNR and SSIM measure different aspects of image quality. While PSNR focuses on pixel-level fidelity, SSIM assesses structural similarity, which can be influenced by various factors. We have added further analysis in the revised paper to explain this trade-off and provide insights for selecting the optimal configuration based on specific application needs.
>
> > Since B controls the brightness of the output image, it is not related to the input images. Consider a case, if I want to reconstruct a bright image (set B=0.8), but with two different input images (one is bright, one is dark), how the result images will be?
>
> Thank you for your question. We have added a discussion on this scenario in the supplementary material, providing examples and analysis. We appreciate your insightful comment, which has helped us enhance the clarity of our work.

---

> ### Author Response · Authors · 2024-11-24
> **Looking Forward to Further Discussion to Reviewer G7fd**
>
> Dear Reviewer G7fd,
>
> We would like to kindly remind you that in **Appendix Section C** of the revised paper, we have provided new illustrative examples in response to your questions. Specifically, for a single scene, we used extremely low-light and extremely overexposed images as inputs and presented results for brightness prompts ranging from 0.2 to 0.8. We found that the outputs controlled by the prompts effectively adjusted the brightness of the images.
>
> We sincerely hope that this newly added material addresses your concerns.
>
> Thank you for your valuable questions, which have helped make our paper more complete. We look forward to further discussions with you.
>
> Sincerely,

---

> > ### Comment · Reviewer_G7fd · 2024-11-25
> >
> > Thank you for your response.
> >
> > After reviewing the authors' response, I have gained a deeper understanding of this submission, and the contributions of this work are now more evident. However, I still have some concerns regarding the dataset and method:
> >
> > 1. While some existing datasets utilize APS frames from DAVIS346 event cameras as ground truth, the event-based vision community would benefit from higher-quality datasets to drive further advancements. Frames of higher quality should ideally be provided as ground truth to enhance the dataset’s utility.
> >
> > 2. The additional experiments on the brightness prompt B seem to suggest that it has minimal impact on the output images. This raises questions about the practical effectiveness of brightness prompt B and its contribution to the overall approach.

---

> > > ### Author Response · Authors · 2024-11-25
> > > **Gratitude and Looking Forward to Further Discussion with Reviewer G7fd**
> > >
> > > Dear Reviewer G7fd,
> > >
> > > **Thank you for your prompt and constructive response, which we deeply appreciate. We are so happy to see that you have gained a deeper understanding of our work and recognized its contributions as more evident.**
> > >
> > > Please allow us to further address your concerns regarding the limitations of current devices and the effectiveness of Prompt \( B \).
> > >
> > > - Currently, the DVS346 sensor is one of the most widely used devices in the academic community. Alternative sensors, such as those from Prophesee [a], can only output events without providing well-aligned RGB frames. The DVS346 remains a practical choice for building datasets that include both events and RGB frames. *While it exhibits minor noise issues (e.g., 4.2% fixed pattern noise), the majority of ground-truth frames generated using this sensor are of high quality.* Additionally, our dataset contributes to the field through its **scale** and **diversity** in lighting conditions.
> > >
> > > - In the revised paper, **Figure 9** demonstrates the results of using events to adjust brightness for the same scene under extreme low-light and overexposed conditions. Our method successfully restores significant details, such as branches and leaves, which were otherwise lost in the input images. Additionally, Prompt \( B \) effectively controls the brightness of the output images. Further analysis can be found in **Section C** of the supplementary material.
> > >
> > > Once again, we thank you for your valuable time and effort. We hope that our response addresses your concerns. Your feedback has been crucial for improving the quality of our paper.
> > >
> > > Sincerely,
> > > *The Authors*
> > >
> > > [a] https://www.prophesee.ai/event-based-sensors/

---

### Official Review · Reviewer_zZNj · 2024-11-01

**Soundness:** 3
**Presentation:** 3
**Contribution:** 3
**Rating:** 5
**Confidence:** 4

**Summary:**

This paper proposes an image enhancement and brightness adjustment method using SEE-600K, a carefully captured dataset spanning different brightness levels.
The SEE-600K dataset is significantly larger than existing datasets and was captured under diverse lighting conditions, making it well-suited for both low-light enhancement and HDR imaging applications.
The proposed enhancement method uses cross-attention to fuse events and images, while the brightness adjustment method leverages brightness prompts to produce results tailored to different brightness levels.
The proposed approach achieves superior results compared to previous methods.

**Strengths:**

1. The SEE-600K dataset is carefully designed and captured, which is also suitable for other low-light enhancement and HDR reconstruction methods to test their results.
2. The brightness adjustment methods take brightness prompt into consideration, which reduces the difficulty of recovering actual brightness level without prior knowledges.

**Weaknesses:**

1. The results of the proposed method are not good enough for over-exposed areas. Some details are missing in saturated areas, e.g., Figure 19 and Figure 20. They are also not good enough for under-exposed areas, e.g., Figure 5.
2. The results of different methods in Figure 5 are not well-aligned. If these results are from different frames, the comparison may not be fair.

**Questions:**

In Table 2, the proposed method shows the worst results when trained on SED for both high light and normal light but achieves the best results when trained on SEE. Which part of the proposed method contributes to this significant improvement for high light and normal light?
Additionally, I noticed that some methods trained on SED are missing when trained on SEE. What is the reason for removing these methods?

---

> ### Author Response · Authors · 2024-11-20
> **Author Response to Reviewer zZNj**
>
> We sincerely thank you for your insightful comments.
>
> > The results of the proposed method are not good enough for over-exposed areas. Some details are missing in saturated areas, e.g., Figure 19 and Figure 20. They are also not good enough for under-exposed areas, e.g., Figure 5.
>
> Thank you for your suggestion. In some over-exposed regions where there is no event data, information can indeed be lost. The maximum information the network can recover is limited by what the events capture. We acknowledge this limitation. We have included a discussion of this issue in the revised paper.
>
> > The results of different methods in Figure 5 are not well-aligned. If these results are from different frames, the comparison may not be fair.
>
> Thank you for your suggestion.  DCE inherently causes image deformation, which may give the impression of misalignment. In Figures 21 and 22, we demonstrate that each frame is aligned, but DCE can cause the images to expand. We have clarified this point in the revised paper to avoid confusion.
>
> > In Table 2, the proposed method shows the worst results when trained on SED for both high light and normal light but achieves the best results when trained on SEE. Which part of the proposed method contributes to this significant improvement for high light and normal light? Additionally, I noticed that some methods trained on SED are missing when trained on SEE. What is the reason for removing these methods?
>
> Thank you for your in-depth question. The significant improvement of SEE-Net in high-light and low-light conditions results from the combination of our method and the new SEE-600K dataset.
>
> Regarding the missing methods, the DCE method did not converge when trained on SEE-600K, resulting in NAN (Not a Number) errors. For EvLowLight, the large size of the SEE-600K dataset prevented convergence within a reasonable time frame (two weeks). To address this, we downsampled the dataset and obtained results, which we have added to the revised paper. We appreciate your feedback, which has helped us improve the completeness of our comparisons.
>
> Thank you again for your valuable comments. Your insights have been instrumental in enhancing the quality of our work.

---

> > ### Comment · Reviewer_zZNj · 2024-11-25
> >
> > Thank you for your response.
> > I still have two concerns about this work:
> > 1. In Figs. 9, 21, and 22, some details in over-exposed areas are actually recorded by events, while the SEENet cannot recover it.
> > 2. As noted by reviewer aQ36, the SEE dataset has relatively low image quality, which results in the low quality results of SEENet. For instance, SEENet’s results appear more blurry in Figs. 5, 17, 19, 20, and 23 compared to other methods.

---

> > > ### Author Response · Authors · 2024-11-25
> > > **Gratitude for the Reviewer’s Response and Request for Further Clarification**
> > >
> > > Dear Reviewer zZNj,
> > >
> > > Thank you for your timely response. We sincerely apologize for any misunderstandings and have carefully rechecked the figures you mentioned.
> > >
> > > We would like to kindly clarify that in the Appendix, **our model outputs include both (e) and (j) with Prompt = 0.5**. Please note that (j) is not a comparison method.
> > > The actual comparison methods are (f), (g), and (i). We also wish to highlight that the model output in (j) contains rich details derived from the event data.
> > >
> > > At the same time, we acknowledge that as this is the first work to explore brightness adjustment using events, there is room for improvement in how models leverage events. We hope that the SEE-600K dataset can serve as a foundation to inspire future advancements in this direction.
> > >
> > > Thank you for your valuable feedback, and we look forward to further discussions.
> > >
> > > Sincerely,
> > > *The Authors*

---

> ### Author Response · Authors · 2024-11-24
> **Looking Forward to Further Discussion to Reviewer  zZNj**
>
> Dear Reviewer zZNj,
>
> Thank you for your time and thoughtful review. In the revised manuscript, we have included experiments with EvLowLight on the SEE-600K dataset.
>
> We hope these additions address your concerns and look forward to further discussions with you.
>
> Sincerely,
> *The Authors*

---

### Official Review · Reviewer_aQ36 · 2024-11-03

**Soundness:** 2
**Presentation:** 2
**Contribution:** 2
**Rating:** 3
**Confidence:** 5

**Summary:**

This paper introduces SEE-Net, a framework for image brightness adjustment using event cameras across a broad range of lighting conditions. It also presents the SEE-600K dataset, containing event-image pairs under various lighting scenarios. The model employs cross-attention to fuse event and image data, enabling prompt-based brightness control. Experimental results suggest SEE-Net’s improved performance over baseline methods on the SDE and SEE-600K datasets.

**Strengths:**

- The SEE-600K dataset expands upon previous datasets, offering more diverse lighting scenarios, which could be useful for broader experimentation in event-based imaging.

- The lightweight architecture of SEE-Net (1.9M parameters) suggests computational efficiency, which may be beneficial in practical applications.

**Weaknesses:**

- The proposed problem of enhancement with event cameras across  broader brightness is not particularly novel. Prior works on event-based HDR (Cui et al., 2024; Yang et al., 2023; Messikommer et al., 2022) have already explored similar concepts, partially addressing the needs this paper claims as unique. The distinction in this paper’s approach does not clearly add new knowledge to the field.


- Also, the problem’s importance is unclear, especially given that established techniques can already perform exposure adjustments during enhancement. Techniques like [1, 2] allow exposure control with brightness factor as prompts. The paper does not demonstrate how SEE-Net outperforms these approaches when combined with event-based imaging theoretically and empirically.

- The core methodology of using cross-attention to merge event and image data is not new and has been applied extensively in similar tasks [3, 4]. Furthermore, the proposed cross-attention module and prompt mechanism are insufficiently justified. There is no clear rationale for why these choices improve performance over simpler fusion techniques, such as concatenation, or why they surpass existing multi-modal enhancement frameworks. The theoretical foundations for the encoding and decoding processes are limited, leaving the importance of each component unclear.

- The SEE-600K dataset is primarily an expanded version of SDE (Liang et al., 2024), constructed with similar strategies and devices, and addressing a similar problem. Although it extends certain aspects through refined engineering techniques, these modifications alone do not constitute a significant novelty or research contribution.

- The SEE-600K dataset shows quality issues, particularly in the normal-light images. Figures 6 and 12 exhibit noticeable artifacts, such as blurriness (e.g., the tree textures in Row 3 of Figure 12, toy contours in Row 1), saturation (e.g., the toys in Row 1), noise (e.g., grass behind bicycles in Row 4), and other visual defects (e.g., ground in Row 1 of Figure 13). These issues detract from the dataset’s value as a high-standard resource and raise questions about its suitability for rigorous research.


[1] Kindling the darkness: A practical low-light image enhancer, ACM MM, 2019

[2] Learning to See in the Dark, CVPR, 2018

[3] Event-Based Video Frame Interpolation With Cross-Modal Asymmetric Bidirectional Motion Fields, CVPR, 2023

[4] Event-Based Fusion for Motion Deblurring with Cross-modal Attention, ECCV 2022

**Questions:**

1. Why is broader brightness adjustment using event cameras necessary when exposure control can be achieved through established techniques? How does SEE-Net theoretically outperform these approaches?

2. What specific performance gains justify the choice of cross-attention over simpler fusion techniques in the context of this problem?

3. Could the authors provide quantitative metrics or examples to verify the SEE-600K dataset’s consistency and quality, addressing the observed artifacts?

---

> ### Author Response · Authors · 2024-11-21
> **Author Response to Reviewer aQ36 (1/2)**
>
> Dear Reviewer aQ36,
>
> We sincerely appreciate your thoughtful and insightful comments.
>
> > The proposed problem of enhancement with event cameras across  broader brightness is not particularly novel. Prior works on event-based HDR (Cui et al., 2024; Yang et al., 2023; Messikommer et al., 2022) have already explored similar concepts, partially addressing the needs this paper claims as unique. The distinction in this paper’s approach does not clearly add new knowledge to the field.
>
> Thank you for pointing this out. In Lines 52–78 of our paper, we discuss the differences between our work and HDR methods. The main distinctions can be summarized in three key points:
>
> - **Different Objectives:** HDR aims to expand the dynamic range of the output image, whereas our goal is to adjust brightness and recover lost details. In other words, HDR tasks pursue a more ambitious and challenging objective.
> - **Different Dataset Construction:** Since HDR focuses on expanding dynamic range, constructing ground truth for HDR is quite difficult. Previous work [a] constructed HDR datasets by merging nine images with different exposure levels, resulting in only 63 scenes. Similar research [b] produced only 1,000 HDR images.
> - **Different Evaluation Metrics:** HDR methods are evaluated based on their ability to expand dynamic range, while our focus is on brightness adjustment.
>
> In summary, our approach extends the event-guided low-light task to accommodate a wider range of lighting conditions. This objective is smaller in scope compared to HDR **but is more practical**. To thoroughly address the differences with HDR, we have added HDR methods to our comparative experiments for an in-depth discussion.
>
>
> > Also, the problem’s importance is unclear, especially given that established techniques can already perform exposure adjustments during enhancement. Techniques like [1, 2] allow exposure control with brightness factor as prompts. The paper does not demonstrate how SEE-Net outperforms these approaches when combined with event-based imaging theoretically and empirically.
>
> Thank you for your insight. Our research focuses on using events to adjust the brightness of images under a wide range of lighting conditions. This is a novel problem that expands the application scope of event-based imaging. Previous methods have only focused on low-light enhancement.
>
> Imaging challenges in both low-light and high-light conditions are common, which underscores the importance of our research.
>
> Moreover, our method fundamentally differs from references [1, 2]. Firstly, these methods are RGB-based and do not utilize the unique characteristics of the event modality. In terms of technical details:
>
> - [1] "Kindling the Darkness: A Practical Low-Light Image Enhancer": This paper studies low-light image enhancement using Retinex theory. In their network design, they do not introduce brightness factors as prompts to control brightness.
> - [2] "Learning to See in the Dark":
>    - This work focuses on RAW-domain ISP. They introduce an amplification ratio in the network to simulate ISO settings. Note that the purpose of the amplification ratio is to amplify brightness, not to control the output brightness. In other words, controlling the ISO amplification does not necessarily guarantee accurate exposure. Modern cameras have automatic ISO algorithms, yet inaccurate exposures still frequently occur.
>    - Our research problem lies in post-imaging exposure adjustment, that is, brightness adjustment after the ISP process.
>
>       > In our pipeline, the amplification ratio is set externally and is provided as input to the pipeline, akin to the ISO setting in cameras.
>
> Thank you for your suggestion and careful observations. We will include discussions on the relevance of these two works in our paper to highlight the significance of our research problem.
>
>
>
> > The core methodology of using cross-attention to merge event and image data is not new and has been applied extensively in similar tasks [3, 4]. Furthermore, the proposed cross-attention module and prompt mechanism are insufficiently justified. There is no clear rationale for why these choices improve performance over simpler fusion techniques, such as concatenation, or why they surpass existing multi-modal enhancement frameworks. The theoretical foundations for the encoding and decoding processes are limited, leaving the importance of each component unclear.
>
> Thank you for your insightful comments. As you mentioned, the cross-attention mechanism is an important tool in multi-modal fusion, which is why we utilize it in our design. To explore its effectiveness more deeply, we have added ablation experiments to compare the cross-attention mechanism with simpler fusion techniques like concatenation. Your valuable feedback has helped make our paper more robust.

---

> ### Author Response · Authors · 2024-11-21
> **Author Response to Reviewer aQ36 (2/2)**
>
> > The SEE-600K dataset is primarily an expanded version of SDE (Liang et al., 2024), constructed with similar strategies and devices, and addressing a similar problem. Although it extends certain aspects through refined engineering techniques, these modifications alone do not constitute a significant novelty or research contribution.
>
> Thank you for your profound insights. Our study is fundamentally different from SDE in several key aspects:
>
> - **Different Research Problem:** SDE focuses only on low-light scenes, whereas our work considers a broader range of lighting conditions. This expansion increases the applicability of event-based vision in more diverse environments.
> - **Different Research Objective:** Previous methods like SDE simply map low-light images to normal-light scenes, ignoring the continuous distribution of light intensity. This can cause ambiguity during training. In contrast, we introduce prompts to avoid this ambiguity, allowing our method to perform well in both low-light and high-light conditions.
> - **Different Dataset Alignment Method:** Although SDE is based on the DVS346 camera, it aligns multiple videos using image-based methods, which may lead to temporal alignment errors. We design an IMU-based alignment algorithm that achieves millisecond-level accuracy, providing a fundamental difference at the data level.
> - **Different Data Scale and Diversity:** Compared to SDE, our SEE-600K dataset includes more scenes, covers a wider range of lighting conditions, and has a larger data scale.
>
> In summary, the SEE-600K dataset addresses different tasks compared to SDE, supports new training methods, and offers higher alignment accuracy and greater diversity. We hope this clarifies the distinctions and answers your concerns.
>
>
> > The SEE-600K dataset shows quality issues, particularly in the normal-light images. Figures 6 and 12 exhibit noticeable artifacts, such as blurriness (e.g., the tree textures in Row 3 of Figure 12, toy contours in Row 1), saturation (e.g., the toys in Row 1), noise (e.g., grass behind bicycles in Row 4), and other visual defects (e.g., ground in Row 1 of Figure 13). These issues detract from the dataset’s value as a high-standard resource and raise questions about its suitability for rigorous research.
>
> Thank you for your careful observation and insights. Our dataset was captured using the DVS346 sensor. While the APS frames from this sensor do face challenges such as noise and saturation issues, it remains one of the most widely used event cameras in the academic community.
>
> We acknowledge that the camera does have artifacts due to its inherent hardware limitations. However, it sufficiently meets the requirements of our research on brightness adjustment under varying lighting conditions. In early-stage academic research, such limitations are sometimes unavoidable.
>
> Our focus is on lighting conditions. In the future, to reduce artifacts, we plan to use newer sensors with better APS quality. We believe that despite its limitations, the current camera is suitable for our task of brightness adjustment.
>
> For more detailed discussions, please refer to *"Author Response to Reviewer G7fd (2/3)"*
>
> > Why is broader brightness adjustment using event cameras necessary when exposure control can be achieved through established techniques? How does SEE-Net theoretically outperform these approaches?
>
> Thank you for your insightful question. Exposure control is not perfect; overexposure or underexposure can still occur [d, e]. Events provide new information with a higher dynamic range, making exposure adjustment across a wide lighting range possible. Additionally, our method allows for pixel-level brightness adjustment, enabling bidirectional control rather than just unidirectional. This flexibility offers greater freedom for post-processing in imaging.
>
>
> > What specific performance gains justify the choice of cross-attention over simpler fusion techniques in the context of this problem?
>
> Thank you for your valuable insight. We have added new ablation experiments comparing the cross-attention mechanism with simpler fusion techniques to demonstrate the performance gains.
>
> > Could the authors provide quantitative metrics or examples to verify the SEE-600K dataset’s consistency and quality, addressing the observed artifacts?
>
> Thank you for your suggestion. With advances in sensor technology and circuit design, we plan to use new sensors in the future to capture datasets with better APS quality.

---

> ### Author Response · Authors · 2024-11-21
> **Reference**
>
> - [1] Kindling the darkness: A practical low-light image enhancer, ACM MM, 2019
> - [2] Learning to See in the Dark, CVPR, 2018
> - [3] Event-Based Video Frame Interpolation With Cross-Modal Asymmetric Bidirectional Motion Fields, CVPR, 2023
> - [4] Event-Based Fusion for Motion Deblurring with Cross-modal Attention, ECCV 2022
>
> - [a] Nico Messikommer, Stamatios Georgoulis, Daniel Gehrig, Stepan Tulyakov, Julius Erbach, Alfredo Bochicchio, Yuanyou Li, and Davide Scaramuzza. Multi-bracket high dynamic range imaging with event cameras. In Proceedings of the IEEE/CVF conference on computer vision and pattern recognition, pp. 547–557, 2022. 1, 2
> - [b] Mengyao Cui, Zhigang Wang, Dong Wang, Bin Zhao, and Xuelong Li. Color event enhanced single-exposure hdr imaging. In Proceedings of the AAAI Conference on Artificial Intelligence, volume 38, pp. 1399–1407, 2024. 1, 2
> - [c] DAVIS346, https://inivation.com/wp-content/uploads/2021/08/2021-08-iniVation-devices-Specifications.pdf
> - [d] Georgiadou, Elissavet, Evangelos Triantafillou, and Anastasios A. Economides. "A Review of Item Exposure Control Strategies for Computerized Adaptive Testing Developed from 1983 to 2005." Journal of Technology, Learning, and Assessment 5.8 (2007): n8
> - [e] Kim, Joowan, Younggun Cho, and Ayoung Kim. "Exposure control using bayesian optimization based on entropy weighted image gradient." 2018 IEEE International conference on robotics and automation (ICRA). IEEE, 2018.

---

> ### Author Response · Authors · 2024-11-24
> **Looking Forward to Further Discussion to Reviewer aQ36**
>
> Dear Reviewer aQ36,
>
> Thank you for your thorough review. We kindly point out that your suggestions have greatly enhanced the completeness of our paper. We have comprehensively addressed your questions in the revised manuscript.
>
> 1. **Regarding the discussion on HDR**, we have provided a more detailed analysis in Lines 52–78 of the main text, Table 1 in the experimental section, and Appendix Section B.
>
> 2. **Regarding the novelty of our technology**, we have discussed the necessity of using events in Appendix Section D.
>
> 3. **Regarding the importance of the cross-attention mechanism**, we have added Case 4 in Table 3 to validate the concatenation method you suggested. In the initial version, we had validated the fusion method of addition and convolution. These results demonstrate the effectiveness of the cross-attention mechanism.
>
> 4. **Regarding the dataset issues**, we have discussed the characteristics of the DVS346 sensor in Appendix Section A, providing a reference for understanding the advantages and limitations of current sensors.
>
> Thank you for your attention to our paper. We hope these revisions address your concerns, and we look forward to further discussions with you.
>
> Sincerely,

---

### Author Response · Authors · 2024-11-20
**Summary of Official Review**

# Summary of Official Review

We sincerely thank all the reviewers for their appreciation of our paper's contributions and their profound insights. Based on your comments, we have conducted a thorough and comprehensive revision of the manuscript. We are very grateful for your assistance.

We apologize for the delayed response due to the addition of some analytical experiments. Below are our responses and revisions.

First, we would like to reaffirm the contributions of our paper and highlight the reviewers' positive feedback.

### Contributions and Reviewers' Affirmation

| **Strength**  | **Reviewer** | **Official Review**   |
| --- | --- | --- |
| **Novel Research Question and Interesting Dataset - SEE-600K** | aQ36  | "The SEE-600K dataset ... **offering more diverse lighting scenarios**, which **could be useful** for **broader experimentation** in event-based imaging."        |
|        | zZNj  | "The SEE-600K dataset is **carefully designed** and captured."  |
|        |       | "SEE-600K, a carefully captured dataset **spanning different brightness levels**."   |
|        |       | "SEE-600K dataset is **significantly larger** than existing datasets and was captured **under diverse lighting conditions**, making it **well-suited** for..."    |
|        | G7df  | "This paper proposed a dataset that contains images **different light conditions**, which may **contribute to the event-based vision community**."  |
|        |       | "The appendix is **detailed with dataset** samples, additional results, and explanation of the proposed network."       |
|        | UvYW  | "The dataset is the **first event-based dataset** covering a broader luminance range."      |
| **Lightweight and Effective Model - SEE Net**    | aQ36  | "The lightweight architecture of SEE-Net (1.9M parameters) suggests **computational efficiency**, which may be **beneficial in practical applications**."  |
|        | zZNj  | "The brightness adjustment methods take brightness prompt into consideration, which **reduces the difficulty** of recovering actual brightness level without prior knowledges." |
|        | UvYW  | "It proposed a framework **effectively utilizes** events to smoothly **adjust image brightness** through the use of prompts."  |
|        |       | "The proposed method achieves the state-of-the-art performance. ***I like the idea about adjusting the brightness*** of images across a broader range of lighting conditions."  |
| **Extensive Experiments**   | aQ36  | "Experimental **results suggest** SEE-Net’s improved performance over baseline methods on the SDE and SEE-600K datasets."      |
|        | zZNj  | The proposed approach achieves **superior results** compared to previous methods.    |

Next, we address each reviewer's individual questions below. We look forward to engaging in further discussions with you.

---

### Meta-Review · Area_Chair_2rfS · 2024-12-19

**Metareview:**

The paper introduces the SEE-600K dataset and SEE-Net, a lightweight architecture aimed at enhancing event-based imaging across broader brightness ranges. The dataset is designed to expand upon current datasets by offering diverse lighting scenarios, potentially useful for broader experimentation. The network architecture proposes using cross-attention to merge event and image data.

***Strengths:***
- The SEE-600K dataset includes more diverse lighting scenarios, which could benefit the event-based vision community by providing a broader range of test conditions.
- SEE-Net's lightweight design  suggests potential computational efficiency, which is a practical advantage in application scenarios.

***Weaknesses:***
- The proposed enhancements do not sufficiently differentiate from existing works which have already explored similar enhancements.
- The SEE-600K dataset exhibits quality issues, especially in normal-light images, which include noticeable artifacts such as blurriness, saturation, and noise.
- The use of cross-attention and other proposed network components lack sufficient justification.
- The comparisons with existing methods need to be strengthen

Despite the potential practical benefits of a lightweight architecture and the broader range of lighting conditions in the SEE-600K dataset, the lack of clear novelty, unresolved quality issues with the dataset, and insufficient methodological advancements lead to the decision to reject this submission. The authors are encouraged to address these significant concerns in future submissions.

**Additional Comments On Reviewer Discussion:**

After reviewing the authors' rebuttal  some reviewer lowered the score after rebuttal. Then there becomes unanimous negative feedback from all reviewers. The reviewers said that the rebuttal did not adequately address the concerns raised. Many of them share one major concern with the quality of the dataset.

---

### Decision · Program_Chairs · 2025-01-22

Reject